# Design of Inhibitors That Target the Menin–Mixed-Lineage Leukemia Interaction

Moses N. Arthur [1,2,†], Kristeen Bebla [3,4,†], Emmanuel Broni [3,†], Carolyn Ashley [3], Miriam Velazquez [3], Xianin Hua [5], Ravi Radhakrishnan [6,7,8], Samuel K. Kwofie [9,10,*] and Whelton A. Miller III [3,4,7,*]

1   Department of Parasitology, Noguchi Memorial Institute for Medical Research (NMIMR), College of Health Sciences (CHS), University of Ghana, Legon, Accra LG 581, Ghana; marthur3@ur.rochester.edu
2   Biomedical Engineering Department, University of Rochester, Rochester, NY 14627, USA
3   Department of Medicine, Loyola University Medical Center, Loyola University Chicago, Maywood, IL 60153, USA; kbebla@luc.edu (K.B.); ebroni@luc.edu (E.B.); cashley1@luc.edu (C.A.); mvelazquez4@luc.edu (M.V.)
4   Department of Molecular Pharmacology & Neuroscience, Loyola University Medical Center, Loyola University Chicago, Maywood, IL 60153, USA
5   Department of Cancer Biology, Perelman School of Medicine, University of Pennsylvania, Philadelphia, PA 19104, USA; huax@mail.med.upenn.edu
6   Department of Bioengineering, School of Engineering and Applied Science, University of Pennsylvania, Philadelphia, PA 19104, USA; rradhak@seas.upenn.edu
7   Department of Chemical and Biomolecular Engineering, School of Engineering and Applied Science, University of Pennsylvania, Philadelphia, PA 19104, USA
8   Department of Biochemistry and Biophysics, Perelman School of Medicine, University of Pennsylvania, Philadelphia, PA 19104, USA
9   Department of Biomedical Engineering, School of Engineering Sciences, College of Basic & Applied Sciences, University of Ghana, Legon, Accra LG 77, Ghana
10  Department of Biochemistry, Cell and Molecular Biology, West African Centre for Cell Biology of Infectious Pathogens, College of Basic and Applied Sciences, University of Ghana, Accra LG 54, Ghana
*   Correspondence: skkwofie@ug.edu.gh (S.K.K.); wmiller6@luc.edu (W.A.M.III)
†   These authors contributed equally to this work.

**Abstract:** The prognosis of mixed-lineage leukemia (MLL) has remained a significant health concern, especially for infants. The minimal treatments available for this aggressive type of leukemia has been an ongoing problem. Chromosomal translocations of the KMT2A gene are known as MLL, which expresses MLL fusion proteins. A protein called menin is an important oncogenic cofactor for these MLL fusion proteins, thus providing a new avenue for treatments against this subset of acute leukemias. In this study, we report results using the structure-based drug design (SBDD) approach to discover potential novel MLL-mediated leukemia inhibitors from natural products against menin. The three-dimensional (3D) protein model was derived from Protein Databank (Protein ID: 4GQ4), and EasyModeller 4.0 and I-TASSER were used to fix missing residues during rebuilding. Out of the ten protein models generated (five from EasyModeller and I-TASSER each), one model was selected. The selected model demonstrated the most reasonable quality and had 75.5% of residues in the most favored regions, 18.3% of residues in additionally allowed regions, 3.3% of residues in generously allowed regions, and 2.9% of residues in disallowed regions. A ligand library containing 25,131 ligands from a Chinese database was virtually screened using AutoDock Vina, in addition to three known menin inhibitors. The top 10 compounds including ZINC000103526876, ZINC000095913861, ZINC000095912705, ZINC000085530497, ZINC000095912718, ZINC000070451048, ZINC000085530488, ZINC000095912706, ZINC000103580868, and ZINC000103584057 had binding energies of −11.0, −10.7, −10.6, −10.2, −10.2, −9.9, −9.9, −9.9, −9.9, and −9.9 kcal/mol, respectively. To confirm the stability of the menin–ligand complexes and the binding mechanisms, molecular dynamics simulations including molecular mechanics Poisson–Boltzmann surface area (MM/PBSA) computations were performed. The amino acid residues that were found to be potentially crucial in ligand binding included Phe243, Met283, Cys246, Tyr281, Ala247, Ser160, Asn287, Asp185, Ser183, Tyr328, Asn249, His186, Leu182, Ile248, and Pro250. MI-2-2 and PubChem CIDs 71777742 and 36294 were shown to possess anti-menin properties; thus, this justifies a need to experimentally determine the activity of the identified compounds. The compounds identified herein were found to have

good pharmacological profiles and had negligible toxicity. Additionally, these compounds were predicted as antileukemic, antineoplastic, chemopreventive, and apoptotic agents. The 10 natural compounds can be further explored as potential novel agents for the effective treatment of MLL-mediated leukemia.

**Keywords:** menin; mixed-lineage leukemia; molecular dynamics; computer-aided drug design

## 1. Introduction

Chromosomal translocations that involve the KMT2A gene, which is also known as the mixed-lineage leukemia 1 (MLL1) gene, are the source of a portion of cases of acute leukemia [1]. This chromosomal rearrangement causes the production of MLL fusion proteins, also known as MLL-FPs, and causes their proliferation [2]. These translocations are found in around 10% of all acute leukemia cases [3]. Mixed-lineage leukemia (MLL)-mediated leukemia affects both children and adults; however, it has a prevalence of 70 to 80% in infants [4,5]. The prognosis for this disease is poor for infants, especially those with primarily acute lymphoblastic leukemias, which comprise approximately 10% of cases and 2.8% of acute myeloid leukemia cases [3,6]. New medicines are urgently needed for the aggressive kind of leukemia caused by MLL1 gene translocation, which affects 5–10% of patients with acute leukemia [7]. Higher rates of relapse and resistance to chemotherapy are two additional effects of the MLL1 mutation [8]. As a result, the prospects for patients with MLL1 rearrangements continue to be dismal compared to those for patients with other types of leukemia.

Within the context of acute leukemia, the menin protein serves as an essential oncogenic cofactor for MLL-FPs [9]. MLL-FPs are distinguished by an N-terminal of DNA-interacting domains of MLL that, when in frame, fuses with the C-terminal of the fusion partner. It has been found that the part of the MLL N-terminus that binds to menin is very important for leukemogenic transformation [10]. Menin is required for the regulation of MLL target genes including HOXA9 and MEIS1, and acts as a highly specific binding partner of MLL-FPs [11]. These genes have been identified to operate in synergy to ensure the maintenance and survival of leukemic cells [12,13]. In vitro and in vivo experiments have shown that the oncogenic ability of MLL-FPs is abolished when they are unable to bind menin. This has established menin as a critical oncogenic cofactor of MLL-FPs necessary for their leukemogenic activity [7,14]. For MLL-FP leukemogenesis, it is crucial to have the interaction between the MLL amino-terminal sequence and menin [9]. Therefore, the discovery of menin inhibitors will provide an avenue for new therapies for MLL.

Currently, there are few therapies that have been identified for MLL-mediated leukemia that are non-toxic [15]. Menin inhibitors are a promising novel therapy and have been found to be most successful for KMT2Ar and NPM1-mutated acute leukemias. However, it is possible that additional subsets of leukemia respond to these treatments as well [9]. The search for new therapies for acute leukemia is currently underway; no fewer than four menin inhibitors are being used clinically after preclinical trials, two of which are SNDX-5613 and KO-539. These inhibitors are currently being tested to determine the tolerability and response rates in patients [9]. This shows that there is a need to identify menin inhibitors as potential therapeutic solutions to MLL-mediated leukemia.

In an effort to identify natural compounds that can be turned into chemotherapies for the treatment of MLL-mediated leukemia, this study aimed to identify menin inhibitors from the Traditional Chinese Medicine (TCM) database [16] via docking. In a previous study that also employed both in silico and in vitro methods, 5 out of 74 compounds tested were found to be potential inhibitors of menin–MLL, with DCZ_M123 having the greatest inhibitory efficacy with an $IC_{50}$ of $4.71 \pm 0.12 \mu M$ [8]. The literature has suggested that menin is a promising target; therefore, research on finding novel drugs to fight against MLL-mediated leukemia is still needed [17,18].

Computer-aided drug design has become one of the most cost-effective and reliable approaches to discovering novel therapies [19]. As previously mentioned, virtual screens of known inhibitors targeting the menin–MLL interface have successfully been performed. However, more research on novel leads is vital to combating MLL-mediated acute leukemia. The purpose of this research is to use computational methods to shortlist potential menin inhibitors derived from natural products. The natural product library of 25,131 compounds was docked against menin using AutoDock Vina [20]. The potential lead compounds were then evaluated to determine if they are anti-cancer drugs using Way2Drug [21]. Then, molecular dynamics (MD) simulations were used to further evaluate the stability of top compounds. In addition, the study compared the binding affinities of known inhibitors to those of potential lead compounds.

## 2. Materials and Methods

This study employed several computational techniques including molecular docking, molecular dynamics simulations, and molecular mechanics Poisson-Boltzmann surface area calculations to identify potential menin inhibitors (Figure 1). Prior to the docking stage, the ability of AutoDock Vina to effectively predict menin binders was evaluated. The shortlisted compounds were further subjected to ADMET screening and biological activity predictions (Figure 1).

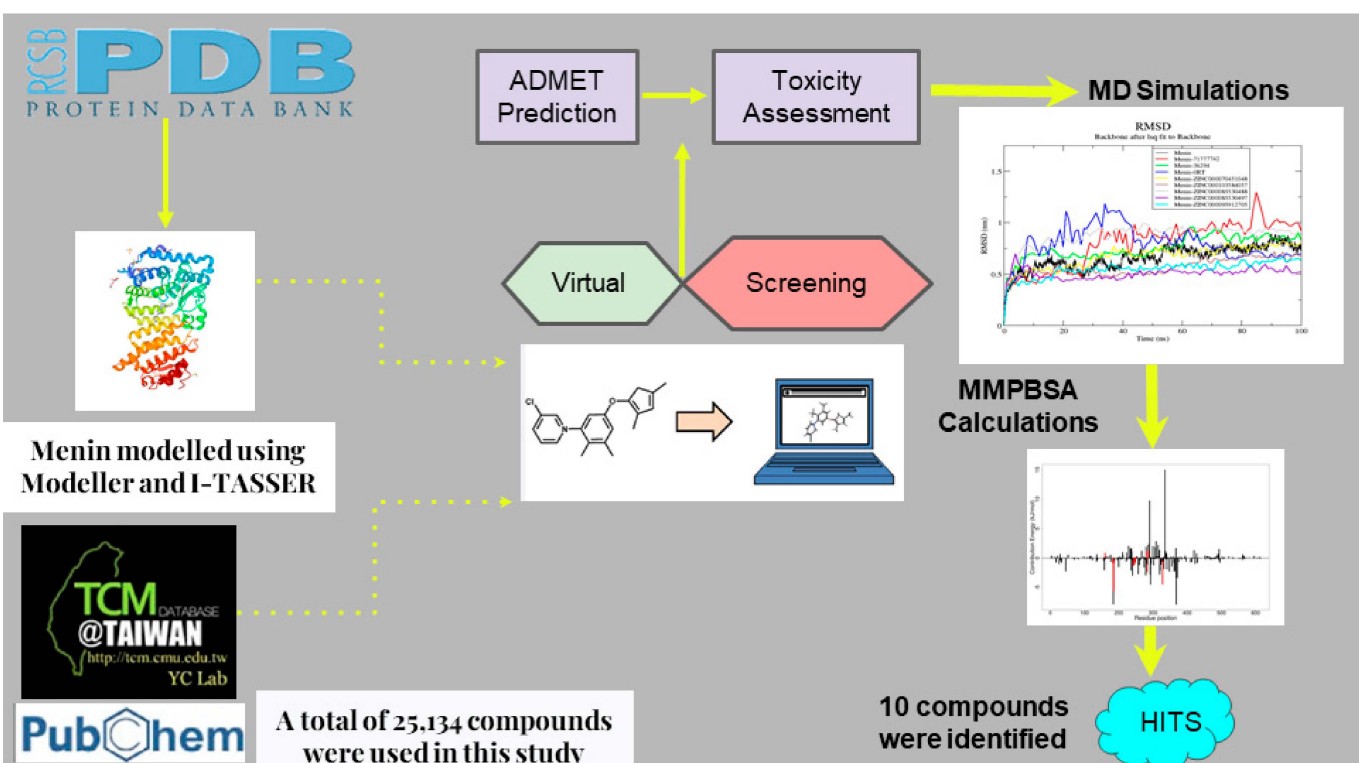

**Figure 1.** Diagram depicting the methods used in this study to predict compounds with the ability to inhibit the menin protein. Natural compounds from the TCM database, and three known menin inhibitors from PubChem, were virtually screened against the menin protein.

### 2.1. Obtaining Protein Structure and Sequence

The Protein Data Bank (PDB) [22] has a three-dimensional (3D) structure of menin with PDB ID 4GQ4 [11]; however, it has missing amino acid residues. Residues 1, 54–73, 387–398, and 462–547 were observed to be missing in the 4GQ4 structure obtained from the PDB [11]. Therefore the sequence of menin was retrieved from UniProtKB [23] with ID O00255, and the missing residues were fixed using EasyModeller [24–26] and I-TASSER [27].

## 2.2. Rebuilding the Menin Structure

Two freely available protein structure modeling platforms comprising EasyModeller and I-TASSER were employed to predict a reasonably good and complete structure of menin. For each modeling platform, five structures were generated. EasyModeller 4.0 was used to generate five models of menin using the 4GQ4 structure as a template. The five structures were assessed based on their discrete optimized protein energy (DOPE) scores and the structure with the lowest DOPE score was selected.

The complete amino acid sequence of menin retrieved from UniProt was used as input for I-TASSER modeling. Using the 3U84 structure, I-TASSER also predicted five structures of complete menin [27–29]. The best I-TASSER-generated structure was selected based on the C-score. The C-score is the confidence quantitative measure for each model that is produced from I-TASSER. The value is based on the significance of threading template alignments and convergence parameters of structure assembly simulations [27–30]. A higher value of the C-score usually indicates a more reliable protein model and it normally ranges from $-5$ to 2 [31].

## 2.3. Model Quality Assessment

The quality of the top 2 models, 1 from each technique, was evaluated using SAVESv6.0 (https://servicesn.mbi.ucla.edu/SAVES/, accessed on 10 February 2023). SAVESv6.0 is a meta-server that runs quality assessments on protein structures. SAVESv6.0 requires the submission of a protein file exclusively in PDB format. Following the upload of the protein structure, the user proceeds by selecting the "Run Programs" button, after which the relevant evaluation programs are initiated by clicking the respective "Start" buttons. Three programs, namely ERRAT [32], VERIFY [33,34], and PROCHECK [35], that can be run by SAVES v6.0 were considered for the quality assessments. Ramachandran plots were obtained via the PROCHECK program. The structure with the better evaluation outcome was selected as reasonably the best model and used for the study.

## 2.4. Obtaining Compounds

The ligand library was obtained from the TCM database [16], which contained 35,161 traditional Chinese medicine natural products. Two known menin inhibitors (PubChem CIDs: 71777742 and 36294) were obtained via PubChem. Another known menin inhibitor, MI-2-2, was extracted from the co-crystallized menin structure in the PDB database (Protein ID: 4GQ4) [11]. The aim was to perform molecular docking studies on these known inhibitors to compare the results to those from the TCM ligand library to aid in prioritizing compounds as potential treatment options for MLL-mediated leukemia.

## 2.5. Protein and Ligand Library Preparation

The 35,161 ligands obtained from the TCM library [16] were pre-filtered using OSIRIS DataWarrior [36] to select only compounds that had molecular weights between 150 and 600 g/mol, as previously carried out [37–40]. It was also used to remove duplicates. Following the filtering of the library with DataWarrior, the 25,196 compounds that were left were subjected to an energy minimization procedure utilizing the universal force field (UFF) [41,42] and conjugate gradient algorithm in 200 steps. The ligands were then converted into PDBQT format, which is supported by AutoDock Vina [20].

The selected protein model was subjected to energy minimization using two different force fields, the optimized potentials for liquid simulations (OPLS)/ all-atom (AA) and CHARMM36 force fields. A short 20 ns MD simulation was also performed using the 2 force fields. This was performed in order to select the force field that would produce a lower energy and a more stable menin structure. The Xmgrace plotting tool was used to generate the graphs. Also, for the selected structure, clustering analysis via the "gmx cluster" command was used to group conformations from the trajectory based on their structural similarity, using the "gromos" method. The representative structure of the largest cluster was then selected.

To eliminate potential docking simulation interferences, the energy-minimized protein model was checked for the presence of water molecules and ions, and these were removed. After that, PyMOL 1.7.4.5 [43] was used to save the protein structure as a pdb file. The "make macromolecule" option in PyRx [44] was used to convert the protein structures into AutoDock Vina's pdbqt format, which was necessary for the virtual screening.

### 2.6. Virtual Screening

AutoDock Vina [20] via PyRx was used to run the virtual screening. A grid box dimension of 32.879 × 25.923 × 30.494 Å$^3$ and the protein centered at x = 57.954 Å, y = 57.286 Å, and z = 58.794 Å were used in order to cover the 0RT binding site. The protein was kept in a rigid conformation during the docking process while the ligands were treated as flexible entities during the docking simulations, allowing AutoDock Vina to generate up to 9 conformations or poses for each compound. Prior to the screening, the quality of the pose prediction conducted via AutoDock Vina was assessed by comparing the best-ranked pose to that in the experimental situation, as previously carried out [37,45–48]. Four known menin inhibitors comprising MI-2 (0RO), MI-2-2 (0RT), MI-89, and MIV-6 were extracted from structures with the corresponding PDB IDs, 4GQ3, 4GQ4, 6E1A, and 4OG8, and were docked against the menin protein. The top-ranked pose, characterized by the most negative binding energy, was subsequently aligned structurally with the conformation in the crystallographic structure using the rigid module of LS-Align [49]. The alignments were then assessed using the RMSD values.

The three known inhibitors and the filtered ligand library were virtually screened against menin. After virtual screening, compounds with binding energies of less than −9.0 kcal/mol were chosen for further investigation. Previous studies have shown that binding affinities that are −7.0 kcal/mol can distinguish between specific and non-specific protein–ligand bonds [50]. All the compounds that passed the −9.0 kcal/mol threshold were analyzed with PyMOL 1.7.4.5 [43] and the best potential ligands that docked deeply in the binding site were studied further. The shortlisted compounds underwent re-docking against the best structure from the clustering analysis to ascertain their binding affinity to menin.

### 2.7. SwissADME Screening of Ligands

After a successful virtual screening, the pharmacological profiles of the selected ligands were obtained through the use of SwissADME [51]. Lipinski and Veber's rules were used to ascertain whether or not these ligands with the highest binding affinities are drug-like. Toxicity risks including the mutagenicity, tumorigenicity, reproductive effect, and irritancy of these ligands were also analyzed using OSIRIS DataWarrior 5.5.0 [36].

### 2.8. Prediction of Biological Activity of Compounds

Prediction of activity spectra for substances (PASS) was utilized to ascertain the biological activities of the shortlisted compounds [21,52]. Using a Bayesian approach, the PASS system is able to predict the biological activity of each compound using its simplified molecular line entry system (SMILES) format as input [21,52].

### 2.9. Determining Binding Interactions

LigPlot+ v.2.2 [53] was used to analyze the protein–ligand interactions using default parameters. This program showed hydrogen bonding and hydrophobic interactions between the protein and ligands. The menin–ligand complexes post-docking were saved in PDB format and uploaded to LigPlot+ v2.2. The ligands' interactions with menin were then visualized and analyzed.

### 2.10. MD Simulations of Protein–Ligand Complexes

For each system, three independent molecular dynamics (MD) runs were performed. To prepare the ligands for the MD simulations, the topologies of each ligand were generated

using the CHARMM General Force Field (CGenFF) server [54]. GROMACS version 5.1.5 was used to run the MD simulations for the unbound menin and the protein–ligand complexes. Each system was solvated in a cubic box positioned 1.0 nm away from the protein on all sides. Sodium or chlorine ions were used to neutralize each system. After energy minimization, the complexes were equilibrated to achieve optimum conditions of pressure and temperature, and then 100 ns MD simulations were carried out. After the MD simulations, Xmgrace was used to generate graphs for analysis.

### 2.11. MM/PBSA Calculations of Protein–Ligand Complex

The molecular mechanics Poisson–Boltzmann surface area (MM/PBSA) method was utilized to analyze the binding free energies of the menin–ligand complexes [55–57]. This system provides the thermodynamic cycle, which incorporates both the molecular mechanical energies and the continuum solvent representation. The calculations for MM/PBSA were conducted using g_mmpbsa [55]. This program computes the contributing energy components for the complex and individual energies of the residues. Three MM/PBSA calculations were performed on the outputs from the three independent MD runs using the full menin structure in complex with the shortlisted ligands. To plot the computational statistics, the R programing system [58] was employed.

## 3. Results

### 3.1. Primary Structure Analysis

The protein sequence of menin is composed of 615 amino acid residues with UniProtKB ID O00255. ProtParam [59] was used to determine the physical and chemical parameters of menin including its molecular weight and chemical formula. its molecular weight was approximately 67,383.72 Da and its chemical formula is $C_{3003}H_{4709}N_{831}O_{898}S_{16}$.

### 3.2. Remodeling Menin Structure

#### 3.2.1. Protein Structure Identification

The crystal structure of human menin with bound inhibitor MI-2-2 (PDB ID: 4GQ4) was solved using X-ray crystallography with a resolution of 1.27 Å [11]. The leukemogenic effect of the MLL fusion protein can be inhibited using a menin inhibitor, which prevents the menin protein from interacting with MLL [8]. MI-2 is the most potent inhibitor of the menin protein with an $IC_{50}$ of 446 nM. MI-2-2 was developed and the $IC_{50}$ was refined 10-fold. This inhibitor, however, did not show promising results in vivo due to its cellular activity and poor metabolic stability [60]. From the literature, 4GQ4 has been used in other in silico studies with promising results [8]. Thus, the 4GQ4 structure was selected as the main template for the menin protein.

#### 3.2.2. Model Rebuilding Using EasyModeller

To fix missing residues, the protein required additional modeling. EasyModeller 4.0 is a tool used for protein homology modeling and the optimization of the protein models generated [24]. The menin protein (Protein Databank ID: 4GQ4) was solved using X-ray crystallography with a resolution of 1.27 Å. However, due to missing residues in the amino acid sequence in the 4GQ4 structure, the EasyModeller 4.0 program was used to reconstruct the protein structure. Five protein models were generated from EasyModeller with DOPE scores of −68,135.84375, −68,138.74219, −67,765.52344, −68,097.15625, and −68,047.66406, respectively. All five models had a genetic algorithm 341 (GA341) score of 1. The second model (MOD2) with a DOPE score of −68,138.74219 was selected as the best structure among the EasyModeller 4.0-generated models (Figure 2a).

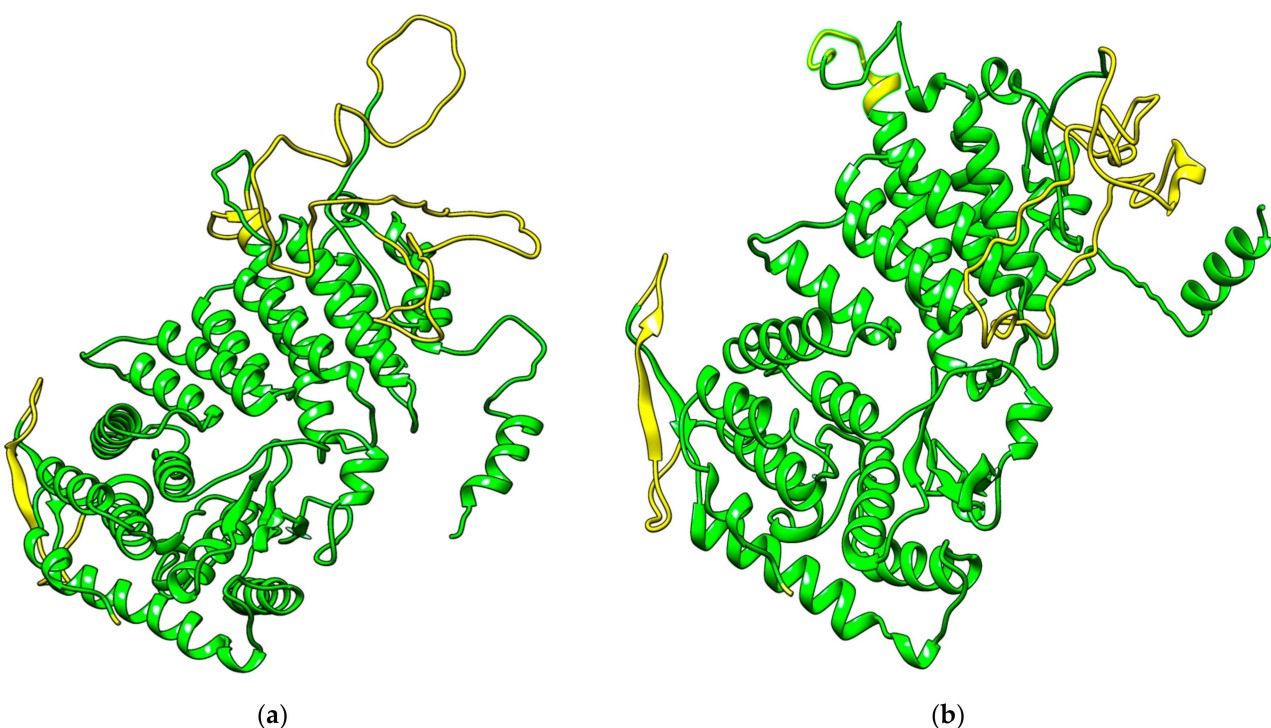

(**a**)                                                   (**b**)

**Figure 2.** Structures of the remodeled menin protein using (**a**) EasyModeller (MOD2) and (**b**) I-TASSER (ITAS1). Yellow-colored protein regions represent missing residues that were fixed during remodeling while the green portions were solved using X-ray crystallography.

### 3.2.3. Structure Prediction Using I-TASSER

Another modeling platform, I-TASSER, was employed to predict reasonable models of the complete menin structure. I-TASSER is a software that predicts protein structures based on structural templates from the protein database [27,30]. I-TASSER [27–30] also predicted five models for the complete structure of menin using PDB ID 3U84 as a template [61]. I-TASSER uses a template-based method to develop protein structures and a program called SPICKER to refine the structural models. Briefly, 3U84 is a menin structure solved at a resolution of 2.5 Å. Five models were predicted using I-TASSER with C-scores of −0.5, −1.2, −0.89, −3.51, and −3.27. The protein structure with the highest C-score of −0.5 (ITAS1) was selected as the most reasonable I-TASSER model for menin (Figure 2b) [30].

### 3.2.4. Protein Model Quality Assessment

Aligning both MOD2 and ITAS1 using PyMOL yielded an RMSD of 0.545 Å, implying that the two structures are not structurally diverse. The structural and stereochemical qualities of ITAS1 and MOD2 were analyzed using SAVES v6.0. ITAS1 had an ERRAT overall quality factor of 88.9447, VERIFY score of 82.44%, and six PROCHECK errors, two warnings, and one pass. The ERRAT plot of ITAS1 demonstrated that amino acid residues 70, 127–130, 133–135, and 202–207 were the most erroneous sections of the protein. For MOD2, the ERRAT score was 70.5479, the VERIFY score was 80.65%, and it had three PROCHECK errors, three warnings, and three passes.

The structured parts of MOD2 and ITAS1 were also analyzed since the disordered regions could influence the quality metrics. It was observed that ITAS1 had an ERRAT overall quality factor of 92.3237 and VERIFY score of 82.06% (pass) while MOD2 had ERRAT and VERIFY scores of 85.1153 and 79.44%, respectively. This showed that the structured regions of ITAS1 had better scores than those of MOD2. ITAS1 was selected as the most reasonable complete menin structure and was used for the study. PROCHECK was employed to compute the Ramachandran plot for ITAS1 to determine the stereochemistry of the protein model based on overall geometry and residue-by-residue geometry [35].

The quality of the protein is based on the percentage of residues in the most favorable, additionally allowed, generously allowed, and disallowed regions, which is shown in the Ramachandran plot [62]. ITAS1 had 391, 95, 17, and 15 residues representing 75.5, 18.3, 3.3, and 2.9% of residues in the most favored regions, additionally allowed regions, generously allowed regions, and disallowed regions, respectively (Figure 3).

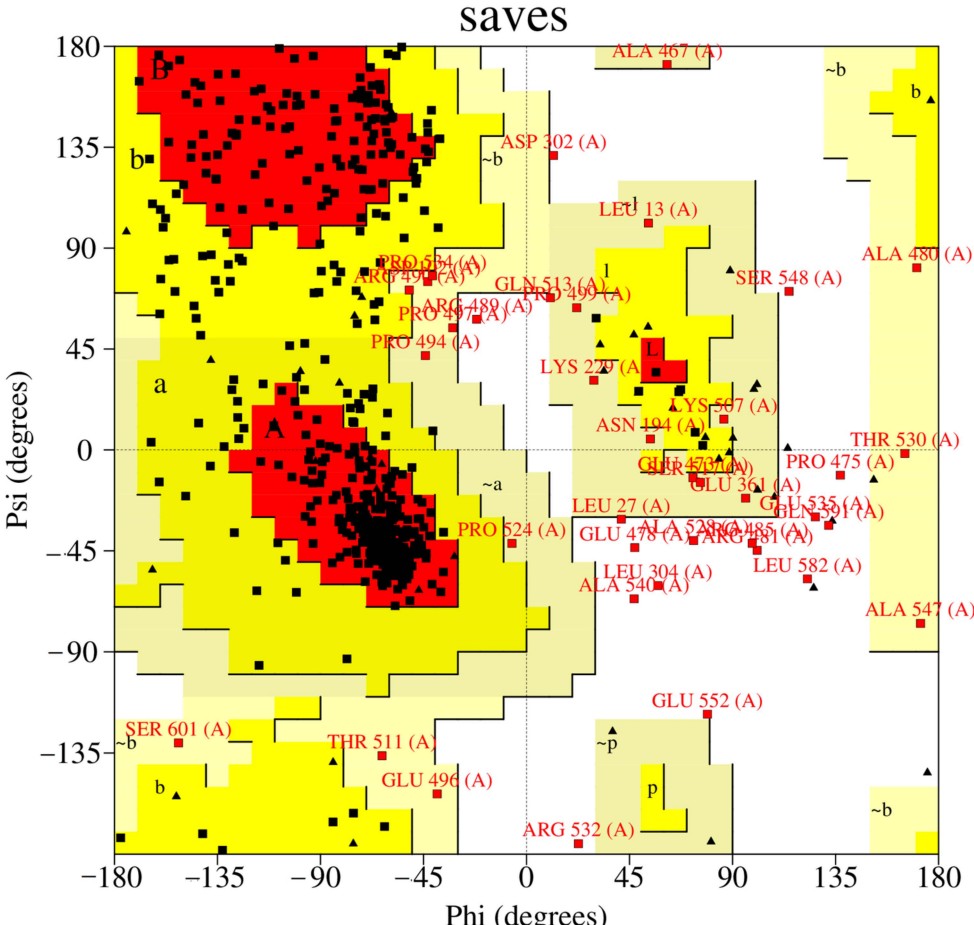

**Figure 3.** Ramachandran plot of the selected menin model (ITAS1) obtained via PROCHECK. The percentages of residues in the most favored regions (red colored; labelled "A", "B", and "L"), additionally allowed regions (yellow colored; labelled "a", "b", "l", and "p"), generously allowed regions (pale yellow; labelled "~a", "~b", "~l", and "~p"), and disallowed regions (white) are 75.5, 18.3, 3.3, and 2.9%, respectively.

### 3.3. Force Field Selection

The CHARMM36 force field was identified to be the optimal fit after subjecting the menin protein to energy minimization and 20 ns MD simulation. OPLS demonstrated a lower energy of −2,784,128.25 kJ/mol in 145 steps while CHARMM36 had a potential energy of −2,706,664.50 kJ/mol in 1090 steps (Supplementary Material Figure S1). However, after the 20 ns MD simulation, the OPLS entire menin structure had an average RMSD of 0.511 ± 0.127 nm while the CHARMM36 entire structure had an average RMSD of 0.382 ± 0.58 nm (Figure S2a), implying that the CHARMM36 structure was more stable. Also, the average Rg value of the CHARMM36 structure (2.829 ± 0.019 nm) was lower than that of OPLS (2.874 ± 0.027 nm) [Figure S2c]. To ensure that the disordered regions did not adversely influence these values, the analyses were also conducted on only the ordered regions of menin. RMSD and Rg analyses specifically focused on the ordered parts of menin

mirrored the trends observed in the analyses of the entire structure (Figure S2b,d). For the ordered regions only, the OPLS and CHARMM36 simulations yielded average RMSDs of $0.47 \pm 0.11$ and $0.36 \pm 0.05$ nm, respectively. Correspondingly, the Rg values for the ordered portions were $2.69 \pm 0.02$ and $2.64 \pm 0.01$ nm for OPLS and CHARMM36 simulations, respectively. The difference in the Rg and RMSD values between the complete protein and the ordered regions was significantly small, that is ~0.05 nm (0.5 Å) for RMSD and ~0.19 nm (1.9 Å) for Rg. Considering the results from the MD analyses, the CHARMM36 force field was selected for the study.

### 3.4. Preparation of Screening Library

A pre-filtered screening library of compounds was produced, which had 25,196 Chinese natural molecules in total from the TCM database. Compounds that have been shown to inhibit the menin–MLL interaction were also identified. MI-2-2 (PubChem CID: 72201086), developed subsequent to MI-2, a potent inhibitor of the menin ($IC_{50}$ of 446 nM), has been reported to exhibit a refined $IC_{50}$ that is tenfold lower than that of MI-2, with a value of 46 nM [63]. MI-2-2 [4-[4-(5,5-dimethyl-4,5-dihydro-1,3-thiazol-2-yl) piperazin-1-yl]-6-(2,2,2-trifluoroethyl) thieno [2,3-d] pyrimidine inhibitor (Code: 0RT)] is shown to inhibit the interaction of menin with MBM1 at an $IC_{50}$ of 46 nM [63]. Due to its inhibitory activity with the menin–MLL interaction, MI-2-2 was one of the known compounds chosen to compare binding affinities against those in the compound library virtually screened. It was extracted from the 4GQ4 complex and saved in sdf format. The compound MIV-6 (PubChem CID: 71777742) is a hydroxy- and aminomethylpiperidine inhibitor which has $IC_{50}$ = 56 nM [8,63]. This class of inhibitors directly interferes with MLL–menin and disrupts the protein–protein interaction. In addition, MIV-6 has been shown to induce differentiation and selectively block the proliferation of MLL cells [64]. An aminoglycoside antibiotic, tobramycin (PubChem CID: 36294), was identified to have Ki = 59.9 μM. Tobramycin has been shown to competitively occupy the active site of MLL-Fps [64]. All ligands used in the study were energy minimized using the UFF under the conjugate gradient algorithm. Prior to virtual screening, the ligands were converted into the pdbqt format using Open Babel [65] in PyRx.

### 3.5. Virtual Screening against ITAS1

AutoDock Vina [20] was used to virtually screen the ligands against the re-modeled menin protein. The program requires a grid box to be pre-calculated, which includes the area of the protein that composes the target binding region. Prior to the docking run, the ability of AutoDock Vina to produce identical poses to experimentally determined conformations was evaluated using LS-Align [49]. The best-ranked poses of each of the four known inhibitors were superimposed onto their co-crystallized poses and the RMSD was determined. The best ranked poses were used for validation since the most energetically favorable conformations of the TCM library will be used for further analysis. For MI-2 (0RO), an RMSD of 1.02 Å was obtained after rigid alignment was performed (Figure S3a). For MI-2-2 (0RT), MI-89, and MIV-6, RMSDs of 2.59, 1.33, and 3.58 Å, respectively, were obtained when the best pose was compared to that of the crystallographic or reference structures (Figure S3b–d). An RMSD equal to or less than 2 Å is deemed very good, while an RMSD between 2 Å and 3 Å is considered acceptable. Conversely, an RMSD exceeding 3 Å is considered suboptimal [66]. Nevertheless, a cut-off value of RMSD < 2 Å is generally recognized as the most effective threshold for validating accurately posed molecules [45,67]. The RMSD values obtained for 0RO and MI-89 show that AutoDock Vina is near-perfect for predicting menin inhibitors, while 0RT's RMSD places AutoDock Vina in the acceptable range. Only one compound's (MIV-6) RMSD value indicates poor pose prediction, suggesting a ~75% chance of AutoDock Vina predicting acceptable or perfect poses for menin binders.

It is worth noting that the second to ninth ranked poses of MI-2-2 (0RT) had RMSD values of 1.95, 2.07, 1.86, 2.41, 2.19, 1.9, 1.52, and 2.03 Å when superimposed onto co-

crystallized MI-2-2 (0RT). This implies that the second to ninth pose predictions were closer to that of the co-crystallized 0RT compared to the best pose. The binding energies of the second to ninth best pose predictions of 0RT were −7.6, −7.4, −7.4, −7.1, −7.1, −7.1, −7.1, and −6.9 kcal/mol. Also, for MIV-6, RMSD values of 3.43, 3.5, 3.1, 3.37, 3.64, 3.33, 3.34, and 3.09 Å were observed when the second to ninth ranked pose predictions were superimposed onto co-crystallized MIV-6.

The protein–ligand interaction profiles of the best poses of MI-2 and MI-89 were also compared to their corresponding complexes obtained from the protein databank. For the 4GQ3 complex, MI-2 interacted with Tyr276 (2.76 Å) and formed hydrophobic contacts with Ser155, Ser178, Glu179, Asp180, His181, Phe238, Cys241, Met278, Tyr319, Met322, Tyr323, and Glu363. It also formed three hydrophobic bonds with three unknown atoms or ions (labeled "UNX"). For the re-docked complex, MI-2 interacted with the menin binding site as though it was a mirror image (enantiomer) of the co-crystallized MI-2. Re-docked MI-2 interacted with similar residues in the binding site, forming two hydrogen bonds with Asp185 (3.07 Å) and Tyr281 (2.88 Å), and hydrogen bonds with Asp158, Ser159, Ser160, Leu182, Ser183, Glu184, Phe243, Cys246, Met283, Asn287, and Tyr328. For the 4GQ3 structure, residues Ser155, Ser178, Glu179, Asp180, Phe238, Cys241, Tyr276, Met278, and Tyr323 map to residues Ser160, Ser183, Glu184, Asp185, Phe243, Cys246, Tyr281, Met283, and Tyr328, respectively, when compared to the canonical sequence of menin. Therefore, re-docked MI-2 interacted with 9 out of the 13 interactions observed between menin and MI-2 in the 4GQ3 structure (Figure 4a). Similar observations were made for MI-89, interacting with 11 common residues, namely Ser155 (Ser160), Glu179 (Glu184), Phe238 (Phe243), Cys241 (Cys246), Ala242 (Ala247), Tyr276 (Tyr281), Tyr319 (Tyr324), Met322 (Met327), Tyr323 (Tyr328), Glu363 (Glu368), and Val367 (Val372) [Figure 4b]. Residues from the 6E1A structure are explicitly listed, with canonical residue information (ITAS1 residues) provided in brackets for clarity. Residue Asp185 of menin has been shown to form an electrostatic bond with Arg12 of MLL1 while residues Cys246, Ala247, and Glu260 of menin interact with Ala11, Pro10, and Ala37 of MLL1 [68]. Also, mutating Glu260 to leucine was reported to cause a loss of the hydrogen bond formed by Glu260 of menin and MLL1 [68]. These results suggest that AutoDock Vina has a high degree of accuracy in predicting nearly optimal poses and interactions of ligands with menin [68].

The compound library containing 25,131 compounds in total and three known menin–MLL interaction inhibitors were successfully screened against the energy-minimized protein. The MI-2-2 (0RT) binding site on the 4GQ4 structure was selected for docking. MI-2-2 is located in a hydrophobic pocket and forms a hydrogen bond with Tyr323 [8,11]. For the canonical menin sequence, Tyr323 on 4GQ4 maps to Tyr318 on the extracted sequence from UniProt (ID: O00255).

Studies have shown that binding affinities that are −7.0 kcal/mol or lower can distinguish between specific and non-specific protein–ligand bonds [50]. Compounds with a binding energy of −9.0 kcal/mol or less were chosen for additional study to further refine the results. In total, 564 compounds passed the criteria. ZINC000103526876 had the highest binding affinity with an energy of −11 kcal/mol (Table 1). Additionally, ZINC000103436976, ZINC000070451007, and ZINC000095913861 had good binding energies of −10.9, −10.7, and −10.7 kcal/mol, respectively (Table 1). The virtual screening results of the known menin–MLL interaction inhibitors were also examined. Tobramycin (CID: 36294) demonstrated a binding energy of −6.7 kcal/mol. MIV-6 (CID: 71777742) was shown to have a binding energy of −6.5 kcal/mol and MI-2-2 (0RT) had a binding energy of −7.8 (Table 1).

The topmost compounds were re-docked against the representative structure of the largest or best cluster after performing clustering analysis on the 20 ns MD system. All the potential lead compounds demonstrated better binding affinity to the menin structure than the known inhibitors (Table 1). The virtual screening results were reviewed using PyMOL 1.7.4.5 [43]. All of the ligands docked deeply into the selected menin binding site. This binding pocket plays a role in the binding of MI-2-2, a potent inhibitor of menin–MLL interaction, and thus may play a pivotal role in developing novel classes of inhibitors.

Disrupting the interaction between MLL-FPs and menin will be essential for the treatment of MLL-mediated leukemia.

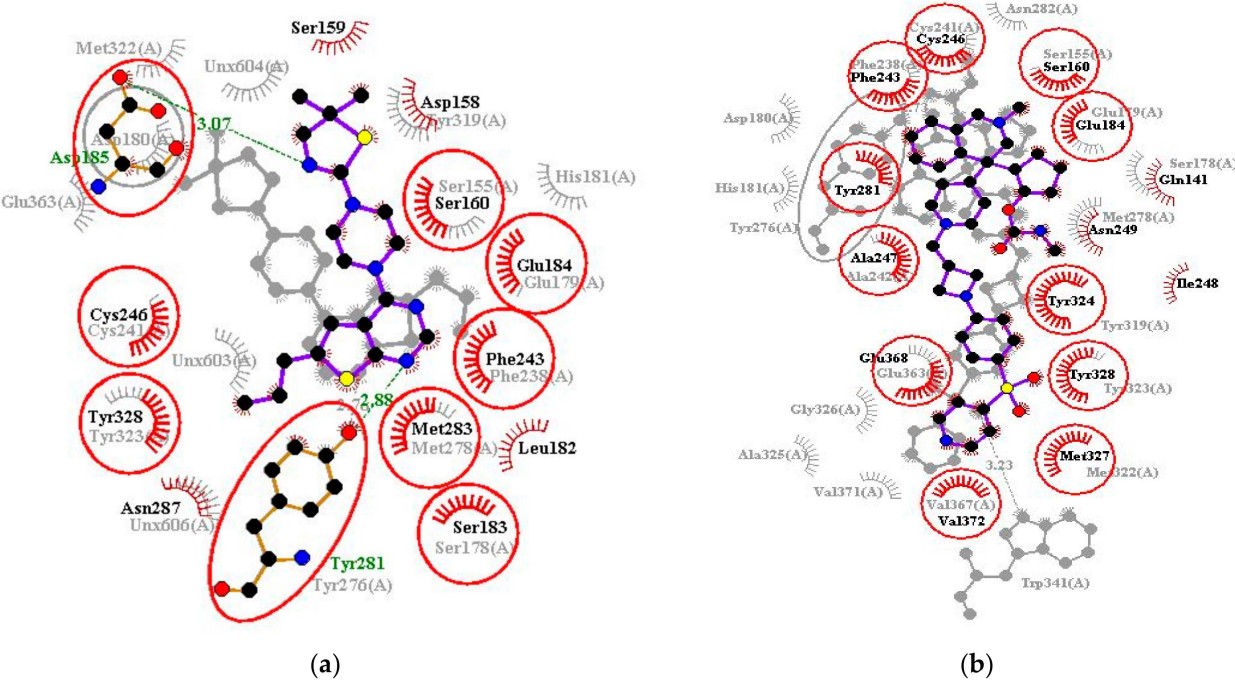

(**a**)　　　　　　　　　　　　　　　(**b**)

**Figure 4.** Superimposition of the "menin-inhibitor" complex and "menin-redocked ligand" complex for (**a**) MI-2 and (**b**) MI-89. Conserved residue interactions are circled in red. The interaction profiles of the crystallographic complexes are transparent while those of the re-docked complexes are rendered clearer and brighter for enhanced visibility. Ligand is colored purple with black dots, red spoked arcs represent hydrophobic bonds, while green dashed lines represent hydrogen bonds. To map residues of the crystallographic structures to the canonical menin structures, add 5 to the residue number of the experimentally determined structures.

**Table 1.** Comparative binding energies and profiles of menin residue interactions with the known inhibitors and the potential lead compounds; docking scores against the representative structure from the largest 20 ns MD cluster.

| Compound | Binding Energy (kcal/mol) | | Interacting Residues | |
|---|---|---|---|---|
| | ITAS1 | Best Cluster | Hydrogen Bonds [Bond Length (Å)] | Hydrophobic Bonds |
| MI-2-2 | −7.8 | −7.5 | Ser160 (2.84) and Ser183 (3.23) | Asp185, His186, Phe243, Cys246, Ala247, Ile248, Asn249, Tyr281, and Met283 |
| 36294 | −6.7 | −6.7 | Asp158 (2.94), Ser160 (3.3), Ser183 (2.79), His186 (2.95), Cys246 (2.9), and Tyr328 (2.83) | Ser159, Leu182, Glu184, Asp185, Phe243, Ala247, and Met283 |
| 71777742 | −6.5 | −6.9 | - | Gln141, Asp158, Ser183, Phe243, Cys246, Ala247, Tyr281, and Met283 |
| ZINC000103526876 | −11.0 | −9.8 | Ser160 (2.92) | Gln141, Asp158, Leu182, Glu184, Asp185, His186, Ala187, His204, Phe243, Cys246, Ala247, Tyr281, Met283, and Glu364 |

**Table 1.** *Cont.*

| Compound | Binding Energy (kcal/mol) | | Interacting Residues | |
|---|---|---|---|---|
| | ITAS1 | Best Cluster | Hydrogen Bonds [Bond Length (Å)] | Hydrophobic Bonds |
| ZINC000095913861 | −10.7 | −10.1 | Ser160 (3.19) and Asn287 (2.7) | Leu182, Ser183, Phe243, Asn249, Pro250, Ser251, Tyr281, Met283, Leu291, and Leu294 |
| ZINC000095912705 | −10.6 | −8.1 | Asn287 (2.99) | Ser160, Leu182, Ser183, Asp185, His186, Ala187, Met233, Phe243, Cys246, Ala247, Ile248, Asn249, Pro250, Tyr281, Met283, and Tyr328 |
| ZINC000085530497 | −10.2 | −9.5 | - | Asp158, Ser160, Phe243, Cys246, Ala247, Ile248, Asn249, Gln265, Tyr281, Met283, and Asn287 |
| ZINC000095912718 | −10.2 | −9.5 | - | Ser160, Phe243, Ala247, Cys246, Pro250, Gln265, Tyr281, Met283, Asn287, Asp290, and Tyr328 |
| ZINC000070451048 | −9.9 | −8.1 | Cys246 (3.02), Ile248 (2.9) and Tyr328 (3.14) | Leu182, Ser183, Glu184, Asp185, His186, Phe243, Ala247, Tyr281, Asn287, Met283, and Asp290 |
| ZINC000085530488 | −9.9 | −8.9 | Ser160 (3.15) and Tyr328 (2.7) | Gln141, Asp158, Asp185, Phe243, Cys246, Ala247, Tyr281, Met283, Asn287, and Tyr324 |
| ZINC000095912706 | −9.9 | −8.1 | Ser160 (3.13 and 3.19), Asn249 (2.97) and er251 (2.93) | Leu182, Ser183, Asp185, His186, Ala187, Phe243, Cys246, Ala247, Ile248, Pro250, Tyr281, Met283, and Asn287 |
| ZINC000103580868 | −9.9 | −9.5 | - | Ser160, Glu184, Phe243, Cys246, Ala247, Ile248, Asn249, Pro250, Ser251, Asp257, Leu262, Gln265, Tyr281, Met283, Asn287, Leu291, and Tyr324 |
| ZINC000103584057 | −9.9 | −8.7 | Arg335 (2.94) | Phe243, Cys246, Tyr281, Met283, Asn287, Asp290, Tyr324, Tyr328, and Glu368 |

### 3.6. ADMET Prediction

Compounds with the highest binding affinity after screening with AutoDock Vina were subjected to profiling using SwissADME [51]. SwissADME is a program that computes pharmacokinetic properties and drug-like nature of compounds to guide drug discovery [51]. The drug-likeness of the ligands with the highest binding affinities was determined based on Lipinski's [69] and Veber's rules. Lipinski's "rule of 5" requires a drug to have a mass less of than 500 g/mol, no more than 10 H-bond acceptors, no more than 5 H-bond donors, and less than 5 octanol–water partition coefficient (log P) values. Veber's rule states that a drug should have a topological polar surface area (TPSA) equal to or less than 140 Å$^2$ and less than 11 rotatable bonds [70].

Although some natural compounds do not adhere to Lipisnki's or Veber's rules, there are current chemotherapy treatments to suggest that they play an important role in the treatment of human ailments [71,72]. Some examples are Taxol, vinblastine, and camptothecin. Additionally, other cancer chemotherapeutic agents come from the marine environment including dolastatins and microbes like bleomycin [71]. It is reasonable to suggest that there are natural biological compounds that would be effective in targeting the binding sites of therapeutic targets. For the TCM ligand library, 360 compounds passed

the Lipinski rule and 375 compounds passed Veber's rule. In total, 295 compounds passed both the Lipinksi and Veber rules and were shortlisted.

OSIRIS DataWarrior 5.0.0 was used to determine the toxicity profiles of the 295 potential lead compounds. Compounds that were high or low in mutagenic and tumorigenic effects were eliminated since the goal of this study was to discover anti-leukemia (anticancer) drugs. In addition, any compounds that had both reproductive and irritancy effects were eliminated. In total, 199 compounds were deemed to be safe and used for further analysis.

### 3.7. Prediction of Biological Activity of Lead Compounds

Prediction of activity spectra of substances (PASS) determines the properties of biologically active substances [49]. The top 10 compounds, ZINC000103526876, ZINC000095913861, ZINC000095912705, ZINC000085530497, ZINC000095912718, ZINC000070451048, ZINC000085530488, ZINC000095912706, ZINC000103580868, and ZINC000103584057, all having binding energies of −9.9 kcal/mol or lower, were further analyzed using PASS. PASS uses the similarity between a query molecule and structures within its training dataset to predict the activity of the query molecule. The probability of a compound to be active (Pa) determines the likelihood of the query compound belonging to the subset of active compounds, while the probability of it being inactive (Pi) suggests the similarity of the query to the inactive subset of the PASS training dataset [73,74]. The probability values (both Pa and Pi) range from 0 to 1, with a higher value suggesting a higher likelihood of activity (Pi) or inactivity (Pi). Generally, compounds with Pa > Pi are deemed likely to demonstrate the particular activity predicted.

Most of the compounds were predicted to have antineoplastic (leukemia, breast, and lung cancer), apoptosis agonist, and chemopreventive properties. The compound ZINC000103526876 with the highest binding affinity (−11 kcal/mol) was predicted as antineoplastic (Pa: 0.864 and Pi: 0.006) and an apoptosis agonist (Pa: 0.655 and Pi: 0.020). ZINC000095913861 with a binding energy of −10.7 kcal/mol was predicted to also have antineoplastic properties (Pa: 0.928 and Pi: 0.005) and to be an apoptosis agonist (Pa: 0.731; Pi: 0.012). Similarly, compounds ZINC000095912705, ZINC000095912706, and ZINC000103584057 were predicted to be antineoplastics (Pa values: 0.927, 0.896, and 0.977; Pi values of 0.005, 0.005, and 0.004, respectively) and apoptosis agonists (Pa values of 0.755, 0.663, and 0.779; Pi values of 0.010, 0.019, and 0.009, respectively). Furthermore, compounds ZINC000085530497, ZINC000095912718, ZINC000085530488, and ZINC000103580868 were predicted to be antineoplastics with Pa values of 0.938, 0.959, 0.931, and 0.812, and corresponding Pi values of 0.004, 0.004, 0.005, and 0.01, respectively. ZINC000085530497 was predicted to be an apoptosis agent (Pa: 0.449 and Pi: 0.052). Antineoplastic agents, also known as anticancer drugs, are a diverse class of drugs. They can be classified as alkylating agents, antimetabolites, and natural products [75]. Cancerous cells, including acute leukemia cells, avoid apoptosis in order to proliferate and resist chemotherapy treatments [6]. Thus, apoptosis agonists are essential to developing novel treatments for acute leukemias including MLL-mediated leukemia.

Multiple compounds were shown to have chemopreventive properties including ZINC000103526876 (Pa: 0.403; Pi: 0.022) and ZINC000095913861 (Pa: 0.538; Pi: 0.011). Chemopreventive medicines serve to prevent cancer by preventing DNA damage or inhibiting the proliferation of pre-cancerous cells that have DNA damage. [76]. Additionally, these agents can induce apoptosis and angiogenesis [77]. There has been extensive research on chemopreventive agents that can be derived from fruits and vegetables. It has been shown that the regular consumption of certain fruits and vegetables can lower the risk of obtaining specific cancers [77–79]. Some of these agents include diallyl sulfide, allicin, capsaicin, and anethol [77]. They have also been discovered to reverse the resistance effects of chemotherapy and radiation in patients who are on these treatments [77].

Some compounds were shown to have antileukemic properties including ZINC000085530497 (Pa: 0.699; Pi: 0.005), ZINC000095912718 (Pa: 0.378; Pi: 0.029), ZINC000085530488

(Pa: 0.562; Pi: 0.010), ZINC000103580868 (Pa: 0.564; Pi: 0.010), and ZINC000103584057 (Pa: 0.709; Pi: 0.005). Discovering new antileukemic drugs is essential to treatment efforts for the subset of patients that harbor resistance to other antileukemic therapies and medications. Thus, there is a growing interest in designing novel drugs that have low clinical toxicity [80].

In addition, most of the compounds were predicted to be antineoplastic agents for other cancers including cervical, lung, and breast cancer. ZINC000103526876 was predicted to be an antineoplastic for breast cancer (Pa: 0.574; Pi: 0.013) and lung cancer (Pa: 0.495; Pi: 0.014) while ZINC000095912705 was predicted as an antineoplastic for lung cancer (Pa: 0.658; Pi: 0.007). ZINC000085530497 was predicted to be an antineoplastic for lung cancer (Pa: 0.708;Pi: 0.010) and ZINC000095912718 was predicted to be so for cervical cancer treatment (Pa: 0.636; Pi: 0.004). ZINC000085530488 was also predicted as an antineoplastic for breast cancer (Pa: 0.734; 0.005) and lung cancer (Pa: 0.657; Pi: 0.007). Additionally, ZINC000095912706 and ZINC000103580868 were predicted as antineoplastics for lung cancer (Pa: 0.566 and 0.569; Pi: 0.010 and 0.01, respectively).

### 3.8. Visualization of the Protein–Ligand Interactions

The hydrogen bond and hydrophobic interactions between menin and the ligands were visualized using LigPlot+ v.2.2 (Figure 5a–d and Figure S4a–i) [53]. A hydrogen bond is formed when a hydrogen atom covalently bonded to an electronegative donor interacts with the lone pair of electrons of an electronegative acceptor [81]. In addition, the shorter a hydrogen bond is, the stronger the bond [82]. 0RT formed hydrogen bonds with Ser183 and Ser160 (3.23 and 2.84 Å, respectively) and hydrophobic interactions with Phe243, Met283, Cys246, Ala247, His186, Asp185, and Tyr281 (Figure 5a and Table 1). Compound 36294 formed six hydrogen bonds with Tyr328, Cys246, Ser183, Ser160, Asp158, and His186 (2.83, 2.90, 2.79, 3.30, 2.94, and 2.95 Å, respectively) and hydrophobic interactions with Met283, Phe243, Leu182, Asp185, Glu184, Ser159, and Ala247 (Figure 5b). From the TCM library, compound ZINC000095913861, with a binding energy of −10.7 kcal/mol, had hydrogen bond interactions with Ser160 and Asn287 (3.19 and 2.70 Å, respectively) and hydrophobic interactions with Pro250, Leu294, Leu291, Asn249, Ser251, Met283, Tyr281, Leu182, Ser183, and Phe243 (Figure 5c). ZINC000095912705, with a binding energy of −10.6 kcal/mol, formed a hydrogen bond with Asn287 (2.99 Å) and hydrophobic interactions with His186, Leu182, Ser160, Asp185, Ala187, Met283, Ser183, Phe243, Met233, Cys246, Pro250, Ile248, Asn249, Tyr328, Ala247, and Tyr281 (Figure 5d).

The amino acid residue Phe243 formed interactions with all of the compounds, making it possibly an important residue. Additionally, Met283, Cys246, Tyr281, Ala247, Ser160, Asn287, Asp185, Ser183, Tyr328, Asn249, His186, Leu182, Ile248, and Pro250 were involved in the majority of binding in the active site (Table 1 and Figure 5 and Figure S4a–i). Thus, these residues are possibly essential amino acid residues needed for the binding and stability of ligands. Furthermore, compound 36294 was discovered to have the maximum hydrogen bonds among the known inhibitors with six bonds, and ZINC000095912706 formed three hydrogen bonds, making it the compound from the TCM database that has maximum hydrogen bond interactions with menin.

### 3.9. Molecular Dynamics Simulations

Based on the virtual screening results and analysis, ten compounds were subjected to three independent 100 ns MD simulations to analyze the protein–ligand interactions. Compounds ZINC000103526876, ZINC000095913861, ZINC000095912705, ZINC000085530497, ZINC000095912718, ZINC000070451048, ZINC000085530488, ZINC000095912706, ZINC000 103580868, and ZINC000103584057 were chosen for further analysis. The unbound menin structure was also subjected to MD simulations. After the MD simulations, the radius of gyration (Rg), root mean square deviation (RMSD), and root mean square fluctuation (RMSF) of the ten compounds and unbound menin were analyzed (Table 2). After the MD simulations, snapshots were generated in 25 ns intervals (0, 25, 50, 75, and 100 ns) for each system to analyze ligand binding to menin. Throughout the 100 ns simulations, all three

MD runs for menin–ligand complexes, excluding 36294, 71777742, and ZINC000070451048, maintained stable binding to the 0RT binding site of menin. In the second MD run of the menin–36294 complex, the ligand exhibited instability at the binding site at 75 and 100 ns. In the case of the menin–71777742 complex (MD run 2), the ligand relocated to a different region at 25 ns and dissociated from the protein at 50 ns. At 75 and 100 ns, 71777742 was observed to bind to menin at a site other than the 0RT binding site. In the second MD run of the menin–ZINC000070451048 complex, the ligand remained tightly bound to menin until the end, where it eventually dissociated from the protein. The average RMSD, Rg, RMSF, and number of hydrogen bonds were calculated using values from all three MD runs, except for simulations in which the ligand dissociated (Table 2).

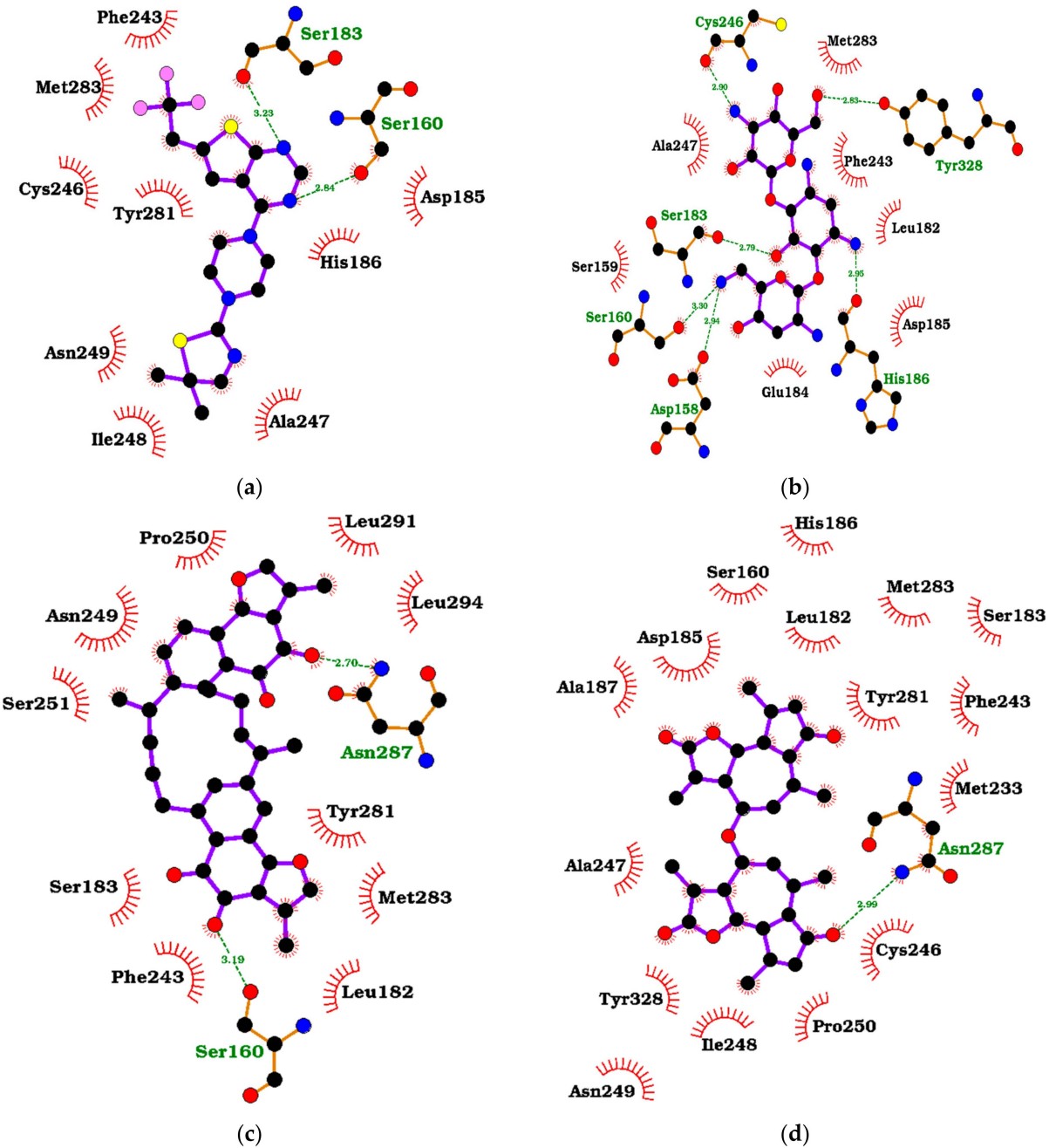

**Figure 5.** Protein–ligand interaction maps of menin in complex with (**a**) MI-2-2 (0RT), (**b**) CID 36294, (**c**) ZINC000095913861, and (**d**) ZINC000095912705. Ligands are colored purple with black dots, red spoked arcs represent hydrophobic bonds, while green dashed lines represent hydrogen bonds.

**Table 2.** Average RMSD, Rg, RMSF, and number of H-Bond values for the unbound menin and menin–ligand complexes after 3 independent 100 ns MD runs. Values are presented as "Value ± Standard deviation" (nm). * denotes values excluded from the average calculation due to an unstable MD run (or ligand dissociation).

| System/Complex | Run | RMSD | | Rg | | RMSF | | H-Bonds |
|---|---|---|---|---|---|---|---|---|
| | | Complete | Ordered | Complete | Ordered | Complete | Ordered | |
| Menin | 1 | 0.66 ± 0.11 | 0.52 ± 0.07 | 2.94 ± 0.02 | 2.71 ± 0.02 | - | - | - |
| | 2 | 0.47 ± 0.08 | 0.45 ± 0.07 | 2.79 ± 0.03 | 2.64 ± 0.03 | - | - | - |
| | 3 | 0.49 ± 0.08 | 0.45 ± 0.06 | 2.84 ± 0.03 | 2.66 ± 0.03 | - | - | - |
| | Avg | 0.54 ± 0.09 | 0.47 ± 0.03 | 2.86 ± 0.06 | 2.67 ± 0.03 | - | - | - |
| Menin–MI-2-2 | 1 | 0.8 ± 0.18 | 0.79 ± 0.2 | 2.87 ± 0.08 | 2.67 ± 0.08 | 0.32 ± 0.34 | 0.29 ± 0.34 | 0.61 ± 0.54 |
| | 2 | 0.79 ± 0.16 | 0.72 ± 0.18 | 2.84 ± 0.05 | 2.63 ± 0.06 | 0.32 ± 0.31 | 0.26 ± 0.31 | 1.11 ± 0.76 |
| | 3 | 0.57 ± 0.07 | 0.53 ± 0.07 | 2.8 ± 0.03 | 2.6 ± 0.04 | 0.26 ± 0.22 | 0.19 ± 0.17 | 0.16 ± 0.37 |
| | Avg | 0.72 ± 0.1 | 0.68 ± 0.11 | 2.84 ± 0.03 | 2.63 ± 0.03 | 0.3 ± 0.03 | 0.25 ± 0.04 | 0.63 ± 0.39 |
| Menin–36294 | 1 | 0.75 ± 0.13 | 0.7 ± 0.1 | 2.84 ± 0.06 | 2.59 ± 0.04 | 0.27 ± 0.22 | 0.2 ± 0.14 | 2.9 ± 1.53 |
| | 2 * | 0.56 ± 0.09 * | 0.46 ± 0.09 * | 2.82 ± 0.04 * | 2.62 ± 0.03 * | 0.26 ± 0.2 * | 0.2 ± 0.16 * | 0.95 ± 1.4 * |
| | 3 | 0.55 ± 0.07 | 0.46 ± 0.05 | 2.83 ± 0.04 | 2.65 ± 0.03 | 0.24 ± 0.2 | 0.19 ± 0.17 | 2.82 ± 1.33 |
| | Avg | 0.65 ± 0.1 | 0.58 ± 0.12 | 2.84 ± 0.01 | 2.62 ± 0.03 | 0.255 ± 0.02 | 0.195 ± 0.01 | 2.86 ± 0.04 |
| Menin–71777742 | 1 | 0.8 ± 0.23 | 0.8 ± 0.26 | 2.86 ± 0.07 | 2.66 ± 0.09 | 0.35 ± 0.44 | 0.33 ± 0.44 | 0.11 ± 0.31 |
| | 2 * | 0.65 ± 0.1 * | 0.59 ± 0.1 * | 2.77 ± 0.03 * | 2.61 ± 0.04 * | 0.25 ± 0.25 * | 0.21 ± 0.24 * | 0.15 ± 0.36 * |
| | 3 | 0.72 ± 0.1 | 0.72 ± 0.1 | 2.74 ± 0.04 | 2.57 ± 0.03 | 0.22 ± 0.19 | 0.18 ± 0.18 | 0.32 ± 0.53 |
| | Avg | 0.76 ± 0.04 | 0.76 ± 0.04 | 2.8 ± 0.06 | 2.62 ± 0.05 | 0.29 ± 0.07 | 0.26 ± 0.08 | 0.22 ± 0.11 |
| Menin–ZINC000103526876 | 1 | 0.71 ± 0.12 | 0.55 ± 0.08 | 2.85 ± 0.04 | 2.63 ± 0.04 | 0.27 ± 0.22 | 0.19 ± 0.17 | 1.56 ± 0.76 |
| | 2 | 0.58 ± 0.11 | 0.47 ± 0.09 | 2.84 ± 0.03 | 2.65 ± 0.04 | 0.3 ± 0.23 | 0.22 ± 0.2 | 1.75 ± 0.91 |
| | 3 | 0.59 ± 0.07 | 0.49 ± 0.07 | 2.85 ± 0.05 | 2.64 ± 0.05 | 0.29 ± 0.25 | 0.22 ± 0.23 | 2.51 ± 0.73 |
| | Avg | 0.63 ± 0.06 | 0.5 ± 0.03 | 2.85 | 2.64 ± 0.01 | 0.29 ± 0.01 | 0.21 ± 0.01 | 1.94 ± 0.41 |
| Menin–ZINC000095913861 | 1 | 0.6 ± 0.1 | 0.57 ± 0.1 | 2.82 ± 0.03 | 2.65 ± 0.03 | 0.25 ± 0.22 | 0.23 ± 0.22 | 1.1 ± 0.72 |
| | 2 | 0.78 ± 0.29 | 0.81 ± 0.35 | 2.81 ± 0.03 | 2.61 ± 0.05 | 0.35 ± 0.47 | 0.35 ± 0.5 | 0.59 ± 0.72 |
| | 3 | 0.68 ± 0.09 | 0.55 ± 0.07 | 2.86 ± 0.05 | 2.64 ± 0.03 | 0.26 ± 0.23 | 0.19 ± 0.18 | 0.91 ± 0.63 |
| | Avg | 0.69 ± 0.07 | 0.64 ± 0.12 | 2.83 ± 0.02 | 2.63 ± 0.02 | 0.29 ± 0.04 | 0.26 ± 0.07 | 0.87 ± 0.21 |
| Menin–ZINC000095912705 | 1 | 0.54 ± 0.09 | 0.45 ± 0.06 | 2.89 ± 0.04 | 2.63 ± 0.03 | 0.24 ± 0.2 | 0.18 ± 0.16 | 1.51 ± 0.57 |
| | 2 | 0.58 ± 0.09 | 0.52 ± 0.08 | 2.9 ± 0.02 | 2.69 ± 0.02 | 0.23 ± 0.18 | 0.18 ± 0.16 | 1.13 ± 0.36 |
| | 3 | 0.74 ± 0.14 | 0.58 ± 0.08 | 2.8 ± 0.02 | 2.58 ± 0.02 | 0.26 ± 0.24 | 0.18 ± 0.15 | 1.09 ± 0.28 |
| | Avg | 0.62 ± 0.09 | 0.52 ± 0.05 | 2.86 ± 0.04 | 2.63 ± 0.04 | 0.24 ± 0.01 | 0.18 ± 0 | 1.24 ± 0.19 |

**Table 2.** *Cont.*

| System/Complex | Run | RMSD | | Rg | | RMSF | | H-Bonds |
|---|---|---|---|---|---|---|---|---|
| | | Complete | Ordered | Complete | Ordered | Complete | Ordered | |
| Menin–ZINC000085530497 | 1 | 0.5 ± 0.07 | 0.44 ± 0.07 | 2.84 ± 0.03 | 2.66 ± 0.03 | 0.21 ± 0.17 | 0.17 ± 0.15 | 0.24 ± 0.45 |
| | 2 | 0.83 ± 0.17 | 0.72 ± 0.18 | 2.92 ± 0.05 | 2.62 ± 0.05 | 0.31 ± 0.32 | 0.25 ± 0.28 | 0.8 ± 0.7 |
| | 3 | 0.75 ± 0.15 | 0.46 ± 0.08 | 2.9 ± 0.03 | 2.63 ± 0.03 | 0.26 ± 0.24 | 0.18 ± 0.18 | 1.51 ± 0.61 |
| | Avg | 0.69 ± 0.14 | 0.54 ± 0.13 | 2.89 ± 0.03 | 2.64 ± 0.02 | 0.26 ± 0.04 | 0.2 ± 0.04 | 0.85 ± 0.52 |
| Menin–ZINC000095912718 | 1 | 0.65 ± 0.1 | 0.56 ± 0.08 | 2.9 ± 0.03 | 2.71 ± 0.03 | 0.26 ± 0.2 | 0.19 ± 0.17 | 0.81 ± 0.69 |
| | 2 | 0.64 ± 0.15 | 0.63 ± 0.15 | 2.82 ± 0.03 | 2.62 ± 0.04 | 0.26 ± 0.28 | 0.22 ± 0.26 | 0.97 ± 0.26 |
| | 3 | 0.64 ± 0.15 | 0.5 ± 0.08 | 2.96 ± 0.06 | 2.73 ± 0.04 | 0.31 ± 0.24 | 0.21 ± 0.18 | 1.43 ± 0.96 |
| | Avg | 0.64 | 0.56 ± 0.05 | 2.89 | 2.69 ± 0.05 | 0.28 ± 0.02 | 0.21 ± 0.01 | 1.07 ± 0.26 |
| Menin–ZINC000070451048 | 1 | 0.67 ± 0.12 | 0.57 ± 0.09 | 2.75 ± 0.04 | 2.6 ± 0.03 | 0.29 ± 0.24 | 0.2 ± 0.2 | 1.15 ± 0.99 |
| | 2 * | 0.64 ± 0.12 * | 0.52 ± 0.15 * | 2.8 ± 0.06 * | 2.65 ± 0.05 * | 0.29 ± 0.27 * | 0.24 ± 0.24 * | 0.72 ± 0.73 * |
| | 3 | 0.7 ± 0.09 | 0.55 ± 0.07 | 2.86 ± 0.04 | 2.7 ± 0.03 | 0.28 ± 0.24 | 0.21 ± 0.2 | 1.11 ± 0.85 |
| | Avg | 0.69 ± 0.02 | 0.56 ± 0.01 | 2.8 ± 0.06 | 2.65 ± 0.05 | 0.29 ± 0.01 | 0.21 ± 0.01 | 1.13 ± 0.02 |
| Menin–ZINC000085530488 | 1 | 0.87 ± 0.13 | 0.63 ± 0.12 | 2.9 ± 0.03 | 2.67 ± 0.03 | 0.27 ± 0.24 | 0.19 ± 0.2 | 0.15 ± 0.41 |
| | 2 | 0.53 ± 0.1 | 0.5 ± 0.11 | 2.81 ± 0.05 | 2.66 ± 0.04 | 0.25 ± 0.24 | 0.2 ± 0.23 | 1.03 ± 0.78 |
| | 3 | 0.63 ± 0.1 | 0.47 ± 0.06 | 2.93 ± 0.04 | 2.67 ± 0.02 | 0.26 ± 0.22 | 0.18 ± 0.16 | 1.01 ± 0.92 |
| | Avg | 0.68 ± 0.14 | 0.53 ± 0.07 | 2.88 ± 0.05 | 2.67 | 0.26 ± 0.01 | 0.19 ± 0.01 | 0.73 ± 0.41 |
| Menin–ZINC000095912706 | 1 | 0.58 ± 0.12 | 0.5 ± 0.11 | 2.86 ± 0.06 | 2.7 ± 0.07 | 0.29 ± 0.28 | 0.24 ± 0.25 | 1.27 ± 0.49 |
| | 2 | 0.61 ± 0.08 | 0.53 ± 0.07 | 2.81 ± 0.04 | 2.64 ± 0.04 | 0.23 ± 0.18 | 0.19 ± 0.15 | 1.78 ± 0.84 |
| | 3 | 0.47 ± 0.05 | 0.41 ± 0.05 | 2.82 ± 0.03 | 2.67 ± 0.03 | 0.22 ± 0.18 | 0.18 ± 0.16 | 1.47 ± 0.57 |
| | Avg | 0.55 ± 0.06 | 0.48 ± 0.05 | 2.83 ± 0.02 | 2.67 ± 0.02 | 0.25 ± 0.03 | 0.2 ± 0.03 | 1.51 ± 0.21 |
| Menin–ZINC000103580868 | 1 | 0.66 ± 0.11 | 0.49 ± 0.07 | 2.83 ± 0.06 | 2.65 ± 0.06 | 0.3 ± 0.28 | 0.21 ± 0.2 | 0.17 ± 0.37 |
| | 2 | 0.69 ± 0.11 | 0.7 ± 0.12 | 2.79 ± 0.04 | 2.58 ± 0.04 | 0.27 ± 0.22 | 0.22 ± 0.19 | 0.01 ± 0.1 |
| | 3 | 0.52 ± 0.07 | 0.44 ± 0.05 | 2.85 ± 0.04 | 2.67 ± 0.03 | 0.24 ± 0.17 | 0.18 ± 0.12 | 0.02 ± 0.14 |
| | Avg | 0.62 ± 0.07 | 0.54 ± 0.11 | 2.82 ± 0.02 | 2.63 ± 0.04 | 0.27 ± 0.02 | 0.2 ± 0.02 | 0.07 ± 0.07 |
| Menin–ZINC000103584057 | 1 | 0.56 ± 0.09 | 0.45 ± 0.06 | 2.84 ± 0.04 | 2.64 ± 0.03 | 0.28 ± 0.21 | 0.21 ± 0.19 | 0.2 ± 0.42 |
| | 2 | 0.56 ± 0.08 | 0.5 ± 0.08 | 2.79 ± 0.04 | 2.62 ± 0.03 | 0.25 ± 0.2 | 0.2 ± 0.17 | 0.19 ± 0.39 |
| | 3 | 0.54 ± 0.1 | 0.45 ± 0.08 | 2.85 ± 0.03 | 2.62 ± 0.03 | 0.25 ± 0.19 | 0.19 ± 0.14 | 0.66 ± 0.63 |
| | Avg | 0.55 ± 0.01 | 0.47 ± 0.02 | 2.83 ± 0.03 | 2.63 ± 0.01 | 0.26 ± 0.01 | 0.2 ± 0.01 | 0.35 ± 0.22 |

The Rg plots of the unbound menin, three menin-inhibitor complexes, and the ten menin–ligand complexes were generated to determine their relative stability (Figure 6 and Figure S5). All systems including unbound menin were found to remain stable throughout the course of the 100 ns MD simulations. The Rg values for the unbound menin were shown to be the most stable throughout the 100 ns simulation course for all three independent MD runs with averages of 2.86 ± 0.06 and 2.67 ± 0.03 nm for the complete protein and only structured regions of menin, respectively. Generally, all the menin–ligand complexes except ZINC000095912718 and ZINC000085530488 demonstrated overall Rg average values that were less than those of menin, signifying a more compact fold during binding. The radius of gyration of a protein determines the protein structure's compactness [83]. The Rg values of a stable protein will remain similar over time; however, if the protein unfolds, then the values will increase or decrease [83]. The disordered regions contributed to higher Rg values for analyses involving the full protein structure. For Rg analyses involving only structured regions of menin, a ~0.2 nm decrease in the full protein's Rg was observed (Table 2).

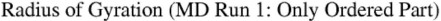

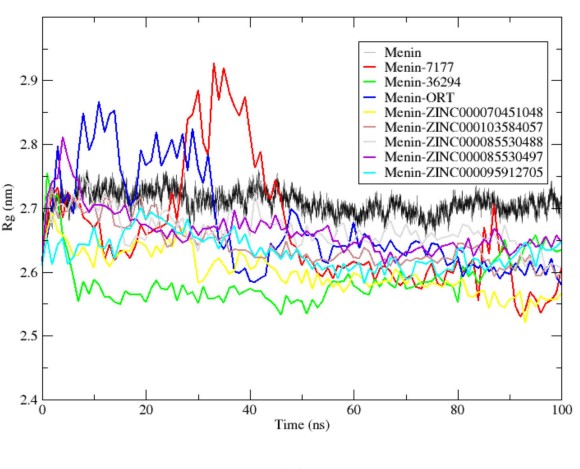

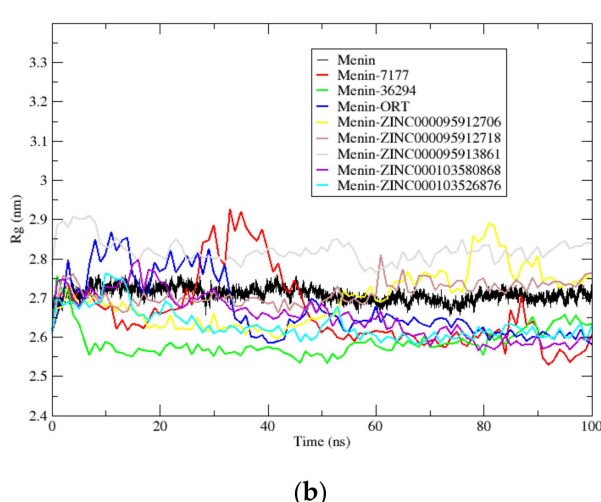

(**a**)                              (**b**)

**Figure 6.** Radius of gyration plots of only the structured regions of the unbound menin protein and the menin–ligand complexes generated using GROMACS after 100 ns MD simulation run 1. The unbound protein is colored black while the known inhibitors, CID 71777742, CID 36294 and 0RT, are colored red, green, and blue, respectively. The unbound menin, and menin complexed with CID 71777742, CID 36294, and 0RT are compared with menin- (**a**) ZINC000070451048, ZINC000103584057, ZINC000085530488, ZINC000085530497, and ZINC000095912705 complexes; and (**b**) ZINC000095912706, ZINC000095912718, ZINC000095913861, ZINC000103580868, and ZINC000103526876 complexes.

To determine the stability of the complexes throughout the simulation compared to that of their reference structure, RMSD plots were generated (Figure 7 and Figure S6) [84]. The RMSD graphs demonstrate how the protein structures of each of the complexes during simulations differ from the reference protein structure over time [84]. For the ten menin–ligand complexes and the unbound menin structure, the RMSD plots showed similar trends. From 0 to 10 ns, the RMSD values rose for all ten menin–ligand complexes and the unbound protein. The unbound menin structure had averages of 0.54 ± 0.09 and 0.47 ± 0.03 nm when RMSDs were determined for the complete protein and only structured regions, respectively (Table 2).

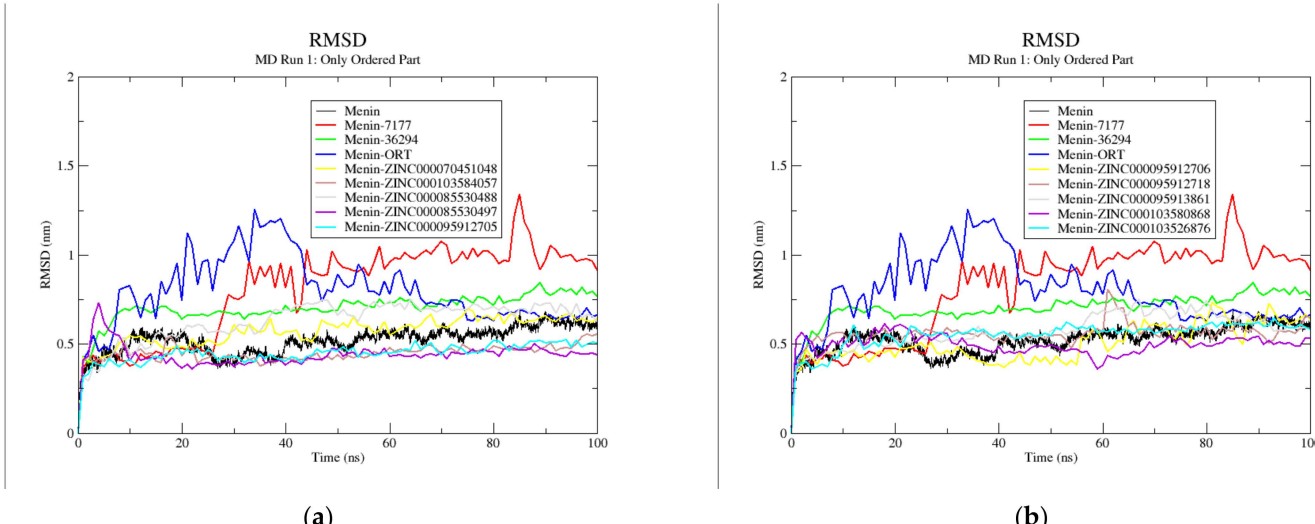

**Figure 7.** Root mean square deviation of only the structured regions of the unbound menin protein and the menin–ligand complexes generated using GROMACS after 100 ns MD simulation run 1. The unbound protein is colored black while the known inhibitors, CID 71777742, CID 36294, and 0RT, are colored red, green and blue, respectively. The unbound menin, and menin complexed with CID 71777742, CID 36294, and 0RT are compared with menin- (**a**) ZINC000070451048, ZINC000103584057, ZINC000085530488, ZINC000085530497, and ZINC000095912705 complexes; and (**b**) ZINC000095912706, ZINC000095912718, ZINC000095913861, ZINC000103580868, and ZINC000103526876 complexes.

To identify the menin residues that contribute to protein structure fluctuations, the RMSF plots of the ten menin–ligand complexes and three menin-inhibitor complexes were analyzed. The protein regions that are most flexible were analyzed using the RMSF plots (Figure 8 and Figure S7a,b). The regions in the RMSF plots with lower values represent the amino acids that are possibly involved in binding and catalysis. The thirteen menin–ligand complexes demonstrated a similar degree of fluctuation in the same amino acid residues in all three independent MD runs. Amino acid positions 260–375 demonstrated the least fluctuations across all complexes in the three MD runs. This is not surprising as most of the ligands interacted with several residues in that region, including Leu262, Gln265, Tyr281, Met283, Asn287, Asp290, Leu291, Leu294, Tyr324, Tyr328, Glu364, and Glu368 (Table 1). Regions with relatively high fluctuations were observed at menin residue indexes 55–60, 70–75, 105–110, 130–135, 150–155, 205–225, 255–260, 390–410, 475–540, implying that these regions may not be involved in ligand binding. The missing residues (1, 54–73, 387–398, and 462–547), which mostly appeared to be disordered in the remodeled structure, were observed to be the regions with the highest fluctuations. The average RMSF values of all the residues in each complex were also determined (Table 2).

### 3.10. MM/PBSA Computations

#### 3.10.1. Binding Energies Involved in Menin–Ligand Binding

To compute the binding free energies of the ten menin–ligand complexes and the three known inhibitor complexes, the molecular mechanics Poisson–Boltzmann surface area (MM/PBSA) was used (Figure 9 and Table 3). The free energy-based simulation that MM/PBSA provides has become increasingly better than other methods including molecular docking to compute ligand–protein binding affinities [65]. The binding free energy calculations determine the affinity to the active site of the protein, which is essential for drug discovery [66]. Since the energy values were computed for all three independent MD runs, overall averages of the energy values were determined for each complex. The overall average energy values were calculated using values from all three MD runs, except for simulations in which the ligand dissociated (36294, 71777742, and ZINC000070451048).

To measure the consistency of or variability in the simulation results, the standard deviations of energy values from the independent MD runs were determined (Figure 9 and Table 3). For all experiments, known inhibitor 36294 demonstrated the highest binding free energy of −55.4 kJ/mol while compound ZINC000095913861 demonstrated the lowest binding free energy of −131.1 kJ/mol followed by ZINC000103580868 with −129.6 kJ/mol (Table 3). ZINC000095912705 demonstrated the least variability in binding free energy with a standard deviation of 1.2 kJ/mol, followed by ZINC000103584057 with 2.5 kJ/mol, while ZINC000095912718 and MI-2-2 had the highest variabilities with standard deviations of 29.1 and 24.3 kJ/mol, respectively across all three runs (Figure 9 and Table 3).

**Figure 8.** Root mean square fluctuation plot for MD run 1 of the menin–ligand complexes generated using GROMACS after 100 ns simulation. The known inhibitors, CID 71777742, CID 36294 and 0RT, are colored purple, red, and green, respectively.

**Table 3.** Energetic insights: MM/PBSA calculations of menin–ligand complexes. The energy values are presented in kJ/mol as "Energy value ± Standard deviation". * denotes values excluded from the average calculation due to an unstable MD run (or ligand dissociation).

| Compound | MD Run | vdW | Electrostatic | Polar Solvation | SASA | Binding Energy |
|---|---|---|---|---|---|---|
| MI−2−2 | 1 | −131.5 ± 1.2 | −16.2 ± 1.3 | 82.4 ± 2 | −16.8 ± 0.1 | −82.1 ± 2 |
| | 2 | −123.5 ± 1.7 | −20.7 ± 1.7 | 103.3 ± 3.9 | −15.8 ± 0.2 | −56.8 ± 2.9 |
| | 3 | −157 ± 1.3 | −2.9 ± 0.6 | 62.2 ± 1.3 | −18.5 ± 0.1 | −116.2 ± 1.5 |
| | Avg | −137.3 ± 14.3 | −13.3 ± 7.6 | 82.6 ± 16.7 | −17 ± 1.1 | −85 ± 24.3 |
| 36294 | 1 | −108.3 ± 1.5 | −53.6 ± 2.5 | 129.1 ± 2.7 | −16.3 ± 0.1 | −49.3 ± 1.7 |
| | 2 * | −36.5 ± 3.9 * | −19.4 ± 2.8 * | 50.5 ± 7.5 * | −6.1 ± 0.6 * | −11.9 ± 5.9 * |
| | 3 | −126 ± 1.6 | −61 ± 2.3 | 143.3 ± 3.4 | −17.7 ± 0.1 | −61.4 ± 2.1 |
| | Avg | −117.2 ± 8.9 | −57.3 ± 3.7 | 136.2 ± 7.1 | −17 ± 0.7 | −55.4 ± 6.1 |
| 71777742 | 1 | −169.6 ± 1.8 | −12.9 ± 0.8 | 54.4 ± 1.6 | −20 ± 0.2 | −148.1 ± 1.7 |
| | 2 * | −75.6 ± 4.6 * | −5.5 ± 0.8 * | 46.1 ± 5.3 * | −9.8 ± 0.6 * | −45 ± 6.8 * |
| | 3 | −168.9 ± 2.7 | −20.6 ± 1 | 77.7 ± 2.1 | −19.1 ± 0.2 | −130.7 ± 2.1 |
| | Avg | −169.3 ± 0.4 | −16.8 ± 3.9 | 66.1 ± 11.7 | −19.55 ± 0.5 | −139.4 ± 8.7 |

**Table 3.** *Cont.*

| Compound | MD Run | vdW | Electrostatic | Polar Solvation | SASA | Binding Energy |
|---|---|---|---|---|---|---|
| ZINC000103526876 | 1 | −131.4 ± 1.5 | −37.4 ± 1.8 | 106.3 ± 3.6 | −18.1 ± 0.2 | −80.8 ± 2.7 |
| | 2 | −173.7 ± 2.5 | −34.1 ± 2 | 110.1 ± 3.3 | −22.4 ± 0.2 | −120.1 ± 2.5 |
| | 3 | −149.3 ± 1.3 | −25 ± 1.6 | 99.4 ± 2.2 | −18.8 ± 0.1 | −93.5 ± 1.8 |
| | Avg | −151.5 ± 17.3 | −32.2 ± 5.3 | 105.3 ± 4.4 | −19.7 ± 1.9 | −98.1 ± 16.4 |
| ZINC000095913861 | 1 | −172.8 ± 1.1 | −53.3 ± 1.3 | 109.3 ± 2 | −21.2 ± 0.1 | −138 ± 1.7 |
| | 2 | −163.4 ± 2.6 | −27.6 ± 2.2 | 95.4 ± 4.1 | −19.7 ± 0.3 | −115.5 ± 2.3 |
| | 3 | −170.3 ± 1.4 | −37.8 ± 1.1 | 89 ± 1.3 | −20.9 ± 0.1 | −139.8 ± 1.7 |
| | Avg | −168.8 ± 4 | −39.6 ± 10.6 | 97.9 ± 8.5 | −20.6 ± 0.6 | −131.1 ± 11 |
| ZINC000095912705 | 1 | −149.9 ± 1.2 | −21.1 ± 1 | 90.8 ± 1.8 | −19.1 ± 0.1 | −99.3 ± 1.5 |
| | 2 | −134 ± 1.2 | −22.5 ± 1.1 | 77.8 ± 1.7 | −17.8 ± 0.1 | −96.5 ± 1.5 |
| | 3 | −151 ± 1.4 | −18.4 ± 1 | 91.4 ± 2 | −19.3 ± 0.1 | −97.3 ± 1.6 |
| | Avg | −145 ± 7.8 | −20.7 ± 1.7 | 86.7 ± 6.3 | −18.7 ± 0.7 | −97.7 ± 1.2 |
| ZINC000085530497 | 1 | −171 ± 1 | −2 ± 1 | 80 ± 1.9 | −20.3 ± 0.1 | −113.3 ± 1.5 |
| | 2 | −177.1 ± 1.9 | −28.4 ± 1.8 | 103.1 ± 2.4 | −20.1 ± 0.1 | −122.4 ± 2.2 |
| | 3 | −140.7 ± 1.3 | −46.3 ± 1.2 | 102.3 ± 2.4 | −17.5 ± 0.1 | −102.3 ± 2 |
| | Avg | −162.9 ± 15.9 | −25.6 ± 18.2 | 95.1 ± 10.7 | −19.3 ± 1.2 | −112.7 ± 8.2 |
| ZINC000095912718 | 1 | −163.1 ± 1.8 | −19 ± 1.3 | 131.4 ± 2.3 | −21.4 ± 0.2 | −72.1 ± 1.7 |
| | 2 | −146.1 ± 1.6 | −51.1 ± 1.1 | 72.5 ± 1.4 | −18.5 ± 0.2 | −143.2 ± 2.1 |
| | 3 | −147.2 ± 1.1 | −59.9 ± 1.3 | 122.6 ± 1.8 | −18.9 ± 0.1 | −103.4 ± 1.8 |
| | Avg | −152.1 ± 7.7 | −43.4 ± 17.6 | 108.8 ± 25.9 | −19.6 ± 1.3 | −106.3 ± 29.1 |
| ZINC000070451048 | 1 | −123.1 ± 1.8 | −123 ± 1.7 | 68.3 ± 2.6 | −16.3 ± 0.2 | −85.4 ± 1.9 |
| | 2 * | −95.4 ± 4.8 * | −28.4 ± 2.2 * | 85.4 ± 5.5 * | −13.3 ± 0.6 * | −51.6 ± 4.4 * |
| | 3 | −165.9 ± 2.2 | −18.5 ± 1.7 | 110.7 ± 3.4 | −19.8 ± 0.2 | −93.5 ± 1.6 |
| | Avg | −144.5 ± 21.4 | −70.8 ± 52.3 | 89.5 ± 21.2 | −18.1 ± 1.8 | −89.5 ± 4.1 |
| ZINC000085530488 | 1 | −154.9 ± 1 | −34.2 ± 1.7 | 106.8 ± 2.3 | −19.4 ± 0.1 | −101.6 ± 1.8 |
| | 2 | −173.9 ± 1.3 | −82.1 ± 1.2 | 170.4 ± 1.4 | −19.8 ± 0.1 | −105.3 ± 1.5 |
| | 3 | −147.8 ± 1.4 | −29.1 ± 1.8 | 120.9 ± 2.8 | −18 ± 0.1 | −74 ± 2.3 |
| | Avg | −158.9 ± 11 | −48.4 ± 23.9 | 132.7 ± 27.3 | −19.1 ± 0.8 | −93.6 ± 14 |
| ZINC000095912706 | 1 | −152.8 ± 1.2 | −40.6 ± 1.2 | 115.3 ± 2 | −19.1 ± 0.1 | −97.3 ± 1.5 |
| | 2 | −163.8 ± 1.4 | −31.9 ± 1.3 | 127.3 ± 2.3 | −20.5 ± 0.1 | −88.9 ± 1.9 |
| | 3 | −165 ± 1.5 | −30.4 ± 1.5 | 137.2 ± 3.8 | −21 ± 0.2 | −79 ± 2.1 |
| | Avg | −160.5 ± 5.5 | −34.3 ± 4.5 | 126.6 ± 9 | −20.2 ± 0.8 | −88.4 ± 7.4 |
| ZINC000103580868 | 1 | −172.4 ± 1.9 | −8.6 ± 0.7 | 94.3 ± 3.4 | −21.6 ± 0.2 | −108.3 ± 3.2 |
| | 2 | −177.9 ± 1.6 | −1.8 ± 0.4 | 56.3 ± 1.8 | −21.1 ± 0.2 | −144.6 ± 2 |
| | 3 | −153.6 ± 2.2 | −2.2 ± 0.6 | 39.6 ± 2.3 | −19.7 ± 0.2 | −135.9 ± 2.8 |
| | Avg | −168 ± 10.4 | −4.2 ± 3.1 | 63.4 ± 22.9 | −20.8 ± 0.8 | −129.6 ± 15.5 |
| ZINC000103584057 | 1 | −128.6 ± 0.9 | −1.8 ± 0.9 | 22.3 ± 2 | −15.2 ± 0.1 | −123.3 ± 1.7 |
| | 2 | −165.5 ± 1.5 | −4.3 ± 0.9 | 70.6 ± 1.6 | −18.2 ± 0.1 | −117.4 ± 1.9 |
| | 3 | −175.6 ± 1.9 | −9.8 ± 1.2 | 85.8 ± 1.7 | −19.4 ± 0.1 | −119 ± 1.7 |
| | Avg | −156.6 ± 20.2 | −5.3 ± 3.3 | 59.6 ± 27.1 | −17.6 ± 1.8 | −119.9 ± 2.5 |

Electrostatic and van der Waal's forces are believed to play a role in binding free energies, as determined from earlier studies [56,85]. Of all the complexes, 36294 demonstrated the highest overall average van der Waal's energy of −117.2 kJ/mol and 71777742 had the lowest van der Waal's energy of −169.3 kJ/mol followed by ZINC000095913861, ZINC000103580868, ZINC000085530497, and ZINC000095912706 with values of −168.8, −168, −162.9, and −160.5 kJ/mol, respectively (Table 3).



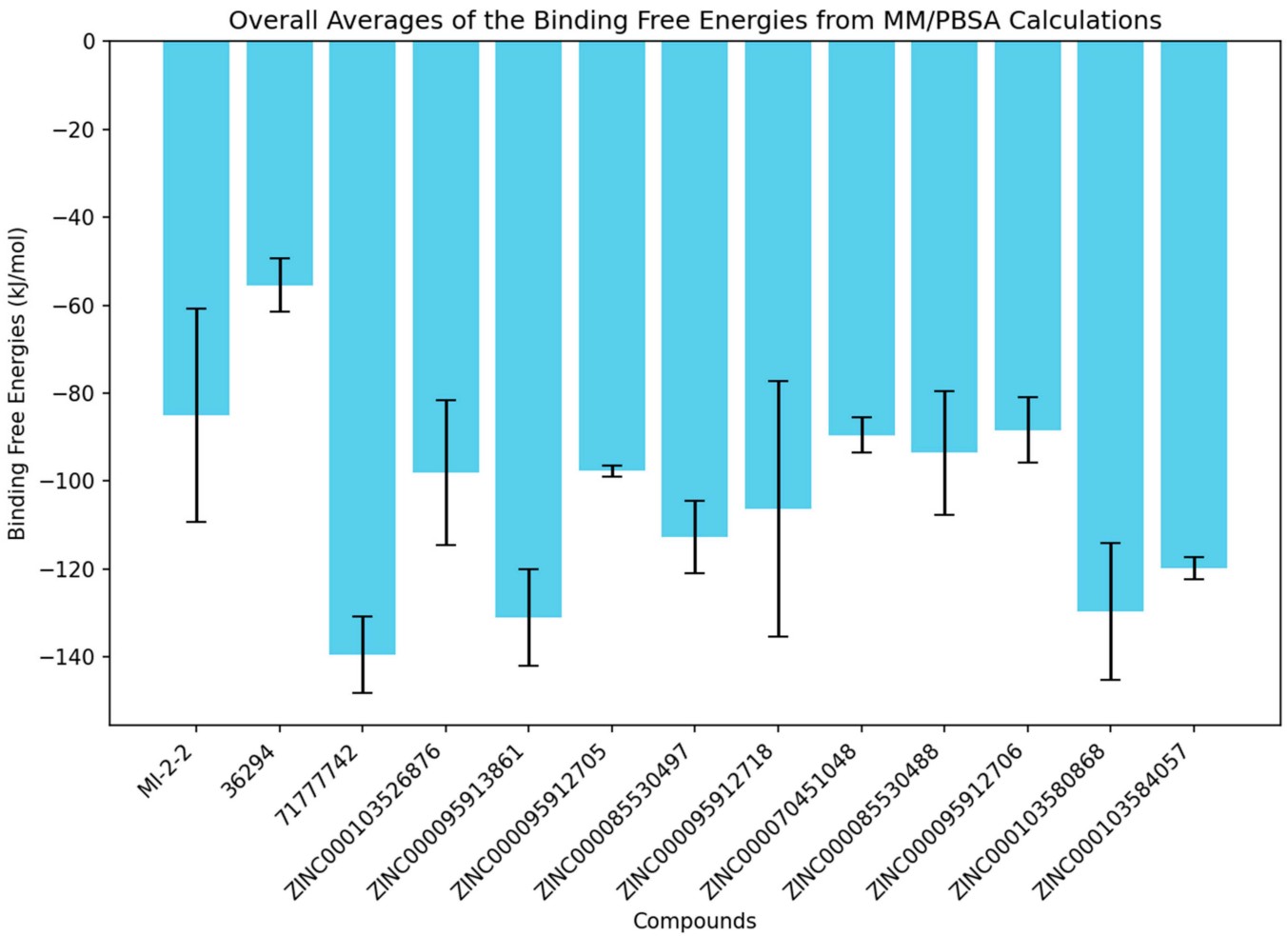

**Figure 9.** Bar plots representing the overall averages of the binding free energies from the 3 independent MD runs computed using the MM/PBSA method. The error bars are the standard deviations calculated from the three binding free energy values of each menin–ligand complex.

3.10.2. Per-Residue Energy Decomposition

To determine the energy contribution of each amino acid residue in the protein structure, the MM/PBSA method was used. This method may aid in the development of inhibitors that target the active site of the protein [86]. A threshold of more than +5 kJ/mol or less than −5 kJ/mol was used and residues with this threshold could be considered critical binding residues [87]. Per-residue energy decomposition calculations were performed for each of the protein–ligand complexes (Figure 10 and Figure S8a–l). The results from the per-residue energy decomposition of only the first independent MD run are discussed herein.

Based on the protein–ligand interaction studies, Phe243, Met283, Cys246, Tyr281, Ala247, Ser160, Asn287, Asp185, Ser183, Tyr328, Asn249, His186, Leu182, Ile248, and Pro250 were all considered potential critical binding residues. For the menin–71777742 complex, Asp185 and Met283 were shown to have energies of less than −5 kJ/mol, which is favorable for ligand binding (Figure S8c). Asp185 and Met283 contributed energies of −8.876 and −5.605 kJ/mol, respectively. The menin–ZINC000070451048 complex had favorable ligand binding to Asp185 (−8.114 kJ/mol) [Figure S8h]. For the menin–36294 complex, Met283 contributed a favorable energy value of −2.8866 kJ/mol (Figure S8b). For the menin–0RT complex, Ser160 contributed a positive energy value of 3.314 and Phe243 contributed a favorable energy value of −3.4485 kJ/mol (Figure S8a). For menin–ZINC000103584057 complex, residues Asp185, Met283, and Tyr328 had favorable energies of −6.123, −5.326,

and −4.465 kJ/mol, respectively (Figure S8l). For menin–ZINC000085530488, a positive energy of 5.075 kJ/mol was observed for Ser160 and negative energies of −5.568, −3.786, and −2.931 kJ/mol were observed for Met283, Asp185, and Tyr328, respectively (Figure S8i). For the menin–ZINC000085530497 complex, Ser160 contributed a positive binding energy of 4.176 kJ/mol and Tyr281 contributed an energy of 3.593 kJ/mol. Asp185, Ala247, and Met283 contributed favorable energies of −8.481, −4.266, and −3.996 kJ/mol, respectively (Figure S8f).

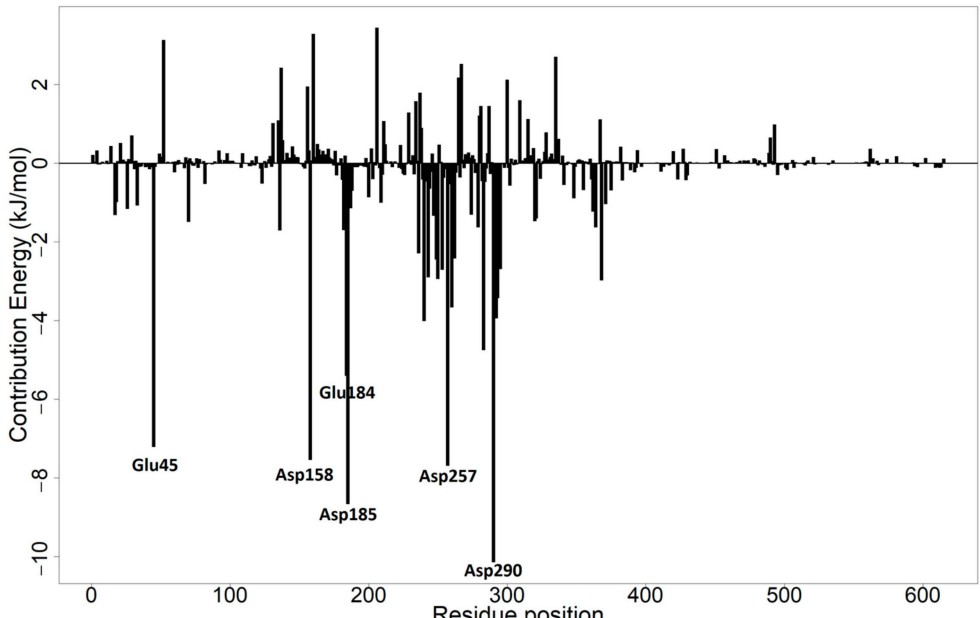

**Figure 10.** Per-residue energy decomposition of the menin–ZINC000095913861 complex. Residues that contributed energies beyond the ±5 kJ/mol threshold have been labeled.

For the menin–ZINC000095912705 complex, Ser160 contributed a positive energy of 3.208 kJ/mol. Asp185, Leu182, Asp185, Phe243, and Met283 contributed energies of −7.131, −6.616, −3.646, −7.131, −3.522, and −6.616 kJ/mol, respectively (Figure S8d). For the menin–ZINC000095912706 complex, Ser160 contributed a positive energy of 3.865 kJ/mol while Asp185, Phe243, Leu182, and Ala247 had favorable energies of −6.842, −5.138, −3.590, and −3.858 kJ/mol, respectively (Figure S8j). For the menin–ZINC000095912718 complex, Met283, Leu182, and Ala247 had favorable energies of −5.350, −3.101, and −3.386 kJ/mol, respectively (Figure S8g). Ser160 contributed a positive energy of 3.287 kJ/mol for the menin–ZINC000095913861 complex (Figure 10). Asp185 and Phe243 also contributed favorable energies of −8.680 and −2.913 kJ/mol. Asn249 and Glu184 also contributed favorable energies of −2.458 and −5.420 kJ/mol, respectively (Figure 10). For the menin–ZINC000103580868 complex, Tyr281 contributed a positive energy of 3.441 kJ/mol while Ala247 and Met283 contributed energies of −3.577 and −4.016 kJ/mol, respectively (Figure S8k). For the menin–ZINC000103526876 complex, Asp185 and Tyr328 contributed favorable energies of −5.600 and −4.4855 kJ/mol, respectively (Figure S8e).

### 3.11. Future Outlook and Implications

This study is one of the few studies that attempts to virtually screen small molecules against the menin structure. The re-modeled protein structure used in this study provides an avenue for the additional discovery of drug-like compounds and better understanding of menin–ligand interactions and promotes catalytic site analysis. Natural products have been under-explored; therefore, this study seeks to show evidence of effective natural products as anti-menin molecules. This study accompanies current endeavors to target the menin protein. These small compounds are crucial to the fight against MLL-mediated

leukemia. An exhaustive pharmacoinformatics approach was used in this study that focused on discovering menin inhibitors targeting the menin–MLL interface. Drug-like molecules were predicted in this study and were fortified with computer system-based antileukemic and antineoplastic activity predictions. MD simulations and MM/PBSA calculations were also employed in this study. For additional depictions of effective future inhibitors, the compounds predicted in the study are recommended to be tested in vivo and in vitro to determine their antineoplastic biological activity. Additionally, this study further calls attention to the repurposing of current molecules as potential inhibitors to the menin protein in a bid to assist the discovery of novel therapies for MLL-mediated leukemia.

## 4. Conclusions

The menin protein is an important oncogenic cofactor for MLL-FPs and a target for MLL-mediated leukemia. Leveraging a multifaceted approach, we employed molecular docking, and in silico physicochemical and pharmacological property predictions to unveil ten potential inhibitors against menin, including noteworthy candidates such as ZINC000103526876, ZINC000095913861, ZINC000095912705, ZINC000085530497, ZINC0000 95912718, ZINC000070451048, ZINC000085530488, ZINC000095912706, ZINC000103580868, and ZINC000103584057. Novel insights into the binding mechanisms and the critical residues of menin were obtained via MD simulations and MM/PSBA calculations. The natural compounds predicted to be potential menin inhibitors were predicted via PASS to have antileukemic, antineoplastic, chemopreventive, and apoptosis agent activities. Additionally, the compounds were predicted to be good drug-like compounds, have good pharmacokinetic properties, and imperceptible toxicity properties. Thus, these lead compounds have the potential to exhibit anti-menin properties during experimental testing. In essence, our comprehensive computational analyses not only contribute to a deeper understanding of menin inhibition but also propose a set of lead compounds with significant potential for anti-menin therapeutic interventions. We anticipate that further experimental investigations will validate and propel these findings toward the development of targeted therapies for MLL-mediated leukemia.

**Supplementary Materials:** The following supporting information can be downloaded at https://www.mdpi.com/article/10.3390/computation12010003/s1. Figure S1: Potential energy plot of menin after energy minimization using OPLS and CHARMM36 force fields. Figure S2: Root mean squared deviation (RMSD) [(a) and (b)] and Radius of gyration (Rg) [(c) and (d)] plots after 20 ns MD simulation of menin using OPLS and CHARMM36 force fields for: (a) and (c) full protein including disordered regions; and (b) and (d) only structured part. Figure S3: Superimposition of the best pose from AutoDock Vina docking to the co-crystalized structures for (a) MI-2 (0RO), (b) MI-2-2 (0RT), (c) MI-89, and (d) MIV-6. For all the superimpositions, the co-crystalized ligands are in red while the docking poses are coloured yellow. Figure S4: Protein-ligand interaction maps of menin in complex with (a) CID 71777742, (b) ZINC000103526876, (c) ZINC000085530497, (d) ZINC000095912718, (e) ZINC000070451048, (f) ZINC000085530488, (g) ZINC000095912706, (h) ZINC000103580868, and (i) ZINC000103584057. Figure S5: Radius of gyration plots of only the structured region of the unbound menin protein and the menin-ligand complexes generated using GROMACS after 100 ns simulation. The Rg plots were generated using: (a) and (b) whole menin structure from MD run 1; (c) and (d) whole menin structure from MD run 2; (e) and (f) only structured regions of menin from MD run 2; (g) and (h) whole menin structure from MD run 3; and (i) and (j) only structured regions of menin from MD run 3. The unbound protein is coloured black while the known inhibitors CID 71777742, CID 36294 and 0RT are coloured red, green, and blue, respectively. Figure S6: Root mean square deviation of the unbound menin protein and the menin-ligand complexes generated using GROMACS after 100 ns simulation. The RMSD plots were generated using: (a) and (b) whole menin structure from MD run 1; (c) and (d) whole menin structure from MD run 2; (e) and (f) only structured regions of menin from MD run 2; (g) and (h) whole menin structure from MD run 3; and (i) and (j) only structured regions of menin from MD run 3. The unbound protein is coloured black while the known inhibitors CID 71777742, CID 36294, and 0RT are coloured red, green and blue, respectively. Figure S7: Root mean square fluctuation plots for

menin-ligand complexes generated using GROMACS after 100 ns simulation for (a) MD run 2 and (b) MD run 3. The known inhibitors CID 71777742, CID 36294 and 0RT are coloured as purple, red, and green, respectively. Figure S8: Per-residue energy decomposition of the menin-ligand complexes: (a) 0RT, (b) 36294, (c) 71777742, (d) ZINC000095912705, (e)ZINC000103526876, (f) ZINC000085530497, (g) ZINC000095912718, (h) ZINC000070451048, (i) ZINC000085530488, (j) ZINC000095912706, (k) ZINC 000103580868, and (l) ZINC000103584057.

**Author Contributions:** S.K.K., W.A.M.III, X.H., R.R. and E.B. conceptualized the study, M.N.A., K.B., E.B., C.A., W.A.M.III and S.K.K. undertook the computational study with contributions from M.V. The first draft was written by M.N.A., K.B., E.B., S.K.K. and W.A.M.III. All authors have read and agreed to the published version of the manuscript.

**Funding:** This research received no external funding.

**Institutional Review Board Statement:** Not applicable.

**Informed Consent Statement:** Not applicable.

**Data Availability Statement:** Data are contained within the article and Supplementary Materials.

**Conflicts of Interest:** The authors declare no conflict of interest.

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
