# Peer review of "Design of Inhibitors That Target the Menin–Mixed-Lineage Leukemia Interaction"

_computation, doi:10.3390/computation12010003_

Round 1

Reviewer 1 Report

Comments and Suggestions for Authors

The authors present a computational exploration of interactions of small molecules with the protein menin, which is a valuable target against diverse types of cancer. Menin is a scaffold protein that links transcription factors to chromatin modifying enzymes and can work as a repressor or an activator of gene transcription depending on context. MLL is a histone methylase and is recruited to DNA via its interaction with menin. This is the interaction that is targeted by the authors, and it has been amply studied by other groups. The work is interesting, ending with a suggestion of novel compounds to test, but I consider that it has a few serious downfalls as described below.

Menin is a large protein (615 amino acids), with one large disordered section according to Uniprot. To solve the lack of coordinates for this region and a couple others in their chosen parent structures (4GQ4 and 3U84), the authors model the missing residues with two approaches (EasyModeller and I-TASSER). As can be seen from the structures in Figure 2, the long yellow loop at the right of the structures has very different predictions, as it should happen given that it is disordered. Actually, if the authors visit the AlphaFold prediction for menin in UNIPROT, these amino acids are identified as very low confidence. Energy minimization and short MD runs will not solve the problem of multiple structures for this region, as conformational changes involving large backbone motions happen in the micro to millisecond time range. Furthermore, the large changes in radius of gyration, rmsd, and rmsf are most likely due to the behavior of this long loop and are not good reporters of menin-inhibitor stability. To ascertain this, the radius of gyration, rmsd, and rmsf should be calculated only for the well-structured amino acids, for which sampling is reasonable at 100ns. Along the same lines, model assessment should also be carried out only for the ordered parts of the structure and compared to that of the parent structure. Oftentimes the quality of a model is inherited from the template. The redeeming fact here is that the binding site for MLL and for the inhibitors is far from the disordered loop. Therefore, I would suggest carrying out a clustering procedure on the structures visited in the 20 ns runs and select the representative structure of the largest cluster as the target for docking.

I find the criterion for force field selection weird; the actual values of the potential energies vary from force field to force field, even for the same structure, so I do not see why this criterion would be useful. The comparison of RMSD and Rg again should be carried out only for the structured part of the protein; conformational sampling of the disordered loop will be different in duplicate runs of the same protein with the same force field.

Regarding binding site characterization, given that there is an MLL-menin complex in the PDB (4GQ6), and this is the interaction that is the target of this project, I do not see the utility of identifying other binding sites. If the authors had stated that they were also interested in allosteric inhibition of MLL binding, that blind search for binding sites would have been useful. Therefore, I would eliminate Table 1.

In section 3.6 the authors state “the MI-2-2 inhibitor was shown to interact with both the high-affinity motif MBM1 and low-affinity motif MBM2 … which are the two menin binding motifs where MLL interacts with menin.” Having checked the cited references, I found that MBM1 and MBM2 exist in MLL, not in menin, so MI-2-2 cannot interact with these motifs. It blocks the binding of these two MLL motifs.

Regarding screening, a control experiment that is missing is the re-docking of MI-2-2 and other known ligands with AutoDock Vina, to validate the docking procedure. The resulting poses should be superimposed on the experimental structures (and shown in a supplementary figure), and the RMSD for the ligand calculated.

Regarding the prediction of biological activity (section 3.9), I strongly suggest presenting these data in a table. In section 3.10, given that all these interactions are already in Table 2, there is no need to write them up again. Instead, I would move the figures in the supplementary material here.

The MD simulations of the menin-ligand complexes, as presented, are not enough to validate binding. Please look up the work of Wonpil Im regarding the use of MD to test the stability of complexes; at least 3 independent 100 ns replicas are needed. As stated above, using Rg as a monitor is not a good idea, as it is extremely likely that all the changes that are seen are due to the large disordered region, and do not reflect what happens with the ligand. The jumps described in the text most likely have to do with this long loop. Instead, the RMSF and RMSD of the ligand in the context of the protein are better stability monitors.

About the free energy calculations, there is not enough information in the Methods section to reproduce them. Were these done using only one run (the complex) or three runs (the complex, free menin, and free ligand)? Which section of the simulation was used for the calculation? Was entropy included? If so, how? It would be good to have error bars in these energies, and it would also be very interesting to compare them qualitatively with those from Vina. In the end, it is the same complex, so one would expect some correlation in the ranking of the complexes (not in the absolute energy values, which are arbitrary). Once these calculations are properly described and there are error bars, I also strongly suggest presenting these data as a Table, including the breakdown into van der Waals and electrostatic interactions. The per-residue energy decomposition can be presented as a figure, and that way it is much easier to follow. By the way, there is no need to have four significant figures in the energies; given the accuracy of the methods, one significant figure is enough.

Comments on the Quality of English Language

The manuscript requires proofreading, as there are missing words now and then, and local grammar issues. Otherwise, it is easy to follow.

Reviewer 2 Report

Comments and Suggestions for Authors

This manuscript represents an excellent and very well presented computational work although in the opinion of this reviewer missing any experimental validation negatively affects the work. In this context PASS predictions are totally useless and only speculative. 

Some specific comments:

line 176: it is not a good choice to delete all the water molecules. Leaving functional water molecules i.e. that ones forming at least 3 Hbonds can lead to better results and better figure out the mechanism of the compounds.

Despite the lots of informations the conclusion are not so strong and in the opinion of this reviewer should be strenghtened.

Overall in consideration of the journal scope the manuscript might be acceptable as such provided that as already mentioned  the missing experimental validation reduces its value . The predicted activity for the main target are not so strong and the value might finally be an artifact (false positive but also false negative).

Reviewer 3 Report

Comments and Suggestions for Authors

The study utilizes a structure-based drug design approach to identify potential inhibitors for menin from natural products. A protein model is created using homology modeling, and a virtual screening of a ligand library is performed. Ten compounds with strong binding energies are identified, and molecular dynamics simulations confirm their stability. This work is somewhat novel, but there are still some problems existed in this study.

1.       The author only used EasyModeller to repair the missing residues, and did not use the homology modeling method to construct a new KMT2A structure. Therefore, the author needs to clarify this.

2.       The author points out that the selected compounds have low toxicity, but in fact, this study did not involve any toxicity calculations or experiments.

3.      4GQ4 is an early resolved protein, and UniProt database has recorded multiple crystal structures of KMT2A (Q03164) protein. The author should try using multiple proteins for screening, compare them, and select compounds that score high in both structures.

Round 2

Reviewer 1 Report

Comments and Suggestions for Authors

The authors have answered all my queries carefully and thoroughly.  Despite that, the manuscript still has some areas that could be improved. 

For example, the figure legend for figure 2 does not include information regarding the color code (what is marked in yellow?).

In the methods section, I urge the authors to read it as if it were not their own, and try to reproduce their calculations with the information given. I still consider that this section is incomplete.

Regarding re-docking of known poses, there are basically two excellent results (0RO and MI-89) and two serious problems (MIV-6 in particular). The problems arise because of the rotation of a dihedral angle, and this could be responding to a bottleneck in the binding site. The docking protocol remains incomplete: was side-chain flexibility allowed during docking? If not, this could explain why some ligands cannot be docked properly. If the ligands do not have enough flexibility, the bent conformations obtained in docking could be inherited from the geometry optimization in gas phase. Was this checked as a possible source of poor agreement with the experimental structure?

In Figure 4 it would be very useful to compare the binding partners of MI-2-2 and MI-89 found by docking with those in the original crystal structure; this lends credibility to the poses generated by VINA, and justifies the detailed description of the interactions.

Table 2 has RMSD, Rg, RMSF, and H-bonds data, but no energies; the header of the table should be corrected accordingly.

From Figure 8 and Table 3 it is clear that systems with very large standard deviations happen because of one run (out of three) that behaves very differently. Do the authors know what transpired in the different runs? Say, for example, run 3 for MI-2-2, or run 2 for 36294. If the simulations are not stable, or the ligand dissociated, they should not be used to estimate these averages. To spot these issues, the radius of gyration of the complex is not useful; maybe the distance of the center of mass of the ligand to an amino acid at the bottom of the binding site would be a useful monitor.

I thank the authors for reminding me that VINA correlates very poorly with MM/PBSA.

In Figure 9, given that the threshold for identification of relevant residues for binding is +/- 5kJ/mol, the important residues could be labeled in the plot.

Comments on the Quality of English Language

I agree with the authors that this version is improved regarding English language. Nevertheless, there is still need of serious proofreading. Auxiliary verbs are missing in some sentences. N-terminal is an adjective, not a noun.

It would be nice to have the definition of Pa and Pi (section 3.7) and the meaning of their numerical values.

Reviewer 3 Report

Comments and Suggestions for Authors

The manuscript can be published.
